# Orientations and water dynamics of photoinduced secondary charge-separated states for magnetoreception by cryptochrome

Misato Hamada[1], Tatsuya Iwata [2✉], Masaaki Fuki[1,3], Hideki Kandori [4,5], Stefan Weber [6✉] & Yasuhiro Kobori [1,3✉]

In the biological magnetic compass, blue-light photoreceptor protein of cryptochrome is thought to conduct the sensing of the Earth's magnetic field by photoinduced sequential long-range charge-separation (CS) through a cascade of tryptophan residues, $W_A(H)$, $W_B(H)$ and $W_C(H)$. Mechanism of generating the weak-field sensitive radical pair (RP) is poorly understood because geometries, electronic couplings and their modulations by molecular motion have not been investigated in the secondary CS states generated prior to the terminal RP states. In this study, water dynamics control of the electronic coupling is revealed to be a key concept for sensing the direction of weak magnetic field. Geometry and exchange coupling (singlet–triplet energy gap: $2J$) of photoinduced *secondary* CS states composed of flavin adenine dinucleotide radical anion ($FAD^{-\bullet}$) and radical cation $W_B(H)^{+\bullet}$ in the cryptochrome DASH from *Xenopus laevis* were clarified by time-resolved electron paramagnetic resonance. We found a time-dependent energetic disorder in $2J$ and was interpreted by a trap CS state capturing one reorientated water molecule at 120 K. Enhanced electron-tunneling by water-libration was revealed for the terminal charge-separation event at elevated temperature. This highlights importance of optimizing the electronic coupling for regulation of the anisotropic RP yield on the possible magnetic compass senses.

[1] Department of Chemistry, Graduate School of Science, Kobe University, 1–1 Rokkodai–cho, Nada–ku, Kobe 657–8501, Japan. [2] Department of Pharmaceutical Sciences, Toho University, Funabashi, Chiba 274–8510, Japan. [3] Molecular Photoscience Research Center, Kobe University, 1–1 Rokkodai–cho, Nada–ku, Kobe 657–8501, Japan. [4] Department of Life Science and Applied Chemistry, Nagoya Institute of Technology, Showa-ku, Nagoya 466-8555, Japan. [5] OptoBioTechnology Research Center, Nagoya Institute of Technology, Showa-ku, Nagoya 466-8555, Japan. [6] Institute of Physical Chemistry, Albert-Ludwigs-Universität Freiburg, 79104 Freiburg, Germany. ✉email: tatsuya.iwata@phar.toho-u.ac.jp; Stefan.Weber@physchem.uni-freiburg.de; ykobori@kitty.kobe-u.ac.jp

Various animals undergo migratory journeys guided by the Earth's magnetic field[1,2]. Several proposals have been put forward on the mechanism of biological magnetoreception[3–6]; however, important aspects are still unclear[7–9]. The perhaps most likely mechanism by which migratory birds sense the magnetic field of the Earth involves the blue-light photoreceptor protein cryptochrome[3,4,10], which was also supposed to be used in signaling in plants and in other animals[8]. Cryptochrome appears to be a quite versatile protein whose role varies depending on the signaling process and the respective organism, be it the entrainment of the circadian clock in vertebrates[11], the regulation of stem elongation in plants, and so on[12,13].

Most members of the cryptochrome protein family exhibit homology in three-dimensional fold, conservation of critical amino acids, and use flavin adenine dinucleotide (FAD) as redox-active cofactor[8,14]. Blue-light induced electron transfer (ET) leads to the generation of long-range charge-separation (CS) state[14]. Several studies highlighted the light-induced ET activity from the protein surface toward the FAD under the participation of redox-active tryptophan residues (Fig. 1)[15–17]. Upon photo-excitation, the FAD in its excited singlet state abstracts an electron from the nearby tryptophan $W_A(H)$ which is part of the so-called conserved "Trp-triad" of $W_A(H)\cdots W_B(H)\cdots W_C(H)$[18,19]. Thus, a short-lived radical pair (RP) composed of the semiquinone anion radical, $FAD^{-\bullet}$, and a tryptophan cation radical, $W(H)^{+\bullet}$, is generated, which represents the primary CS state, i.e. $FAD^{-\bullet}\cdots W_A(H)^{+\bullet}$. Subsequently, quick stepwise sequential electron-hole transfers along the Trp triad take place until the terminal surface-exposed tryptophan ($W_C$) forms a highly separated RP state $FAD^{-\bullet}\cdots W_C(H)^{+\bullet}$. This CS state may further be stabilized by deprotonation to a water molecule forming the RP of $FAD^{-\bullet}\cdots W_C^{\bullet}$[7]. Recently, a fourth tryptophan $W_D(H)$ was demonstrated to be oxidized, thus generating an even more separated RP state $FAD^{-\bullet}\cdots W_D(H)^{+\bullet}$ as the terminal CS state in *Drosophila melanogaster* cryptochrome (*DmCry*) and in pigeon cryptochrome clCRY4[20–22].

In animal cryptochromes the stepwise charge separations proceed from the primary CS state of $FAD^{-\bullet}\cdots W_A(H)^{+\bullet}$, resulting in the terminal CS state[15]. It is essential that the electrostatic stabilizations of the intermediate CS states [primary CS state $FAD^{-\bullet}\cdots W_A(H)^{+\bullet}\cdots W_B(H)\cdots W_C(H)$ and the secondary CS state $FAD^{-\bullet}\cdots W_A(H)\cdots W_B(H)^{+\bullet}\cdots W_C(H)$] are overcome for the ultimate oxidation of the terminal tryptophan to take place. Predictions from the theory have been put forward on how light-induced exergonic oxidization takes place at the terminal tryptophan residue through the Trp triad by stepwise ET[16,23,24]. As an example, it was predicted that the conformation of $W_B(H)$ changes after its oxidation, thus leading to a stabilization of the secondary CS state by the coordination of $W_B(H)^{+\bullet}$ to a threonine residue[23,24]. Water solvation is another origin of the exergonic oxidizations of $W_B(H)$ and $W_C(H)$ because of the higher water accessibility to residues located near the protein surface[16]. If a reorientation of one water molecule is preferential after the charge-separation, no conformation change in $W_B(H)^{+\bullet}$ is required. A recent molecular dynamics (MD) simulation study emphasized the role of forming a hydrogen-bond network involving a captured water molecule located between $W_B(H)$ and $W_C(H)$ in photoactivation by the stepwise CSs in a plant (6–4) photolyase[25].

An advantageous point of utilizing the photoreceptor protein would be the strict controls of distances and orientations (i.e., electronic couplings) between the chromophore and residues for the terminal CS. Moreover, the water fluctuation can be proposed to be a key concept of generating the weak field-sensitive RP. Such solvent dynamics may be a promising perspective to control the electronic couplings applicable to energy conversions with sensing the Earth magnetic field. However, characterization of the electronic coupling has been a challenging task with tracking the time-courses of the intermediate geometries and of the solvation dynamics[26]. Thus, there is no experimental manifestation how the water dynamics can control the electronic coupling in the stepwise CS sequences for the field-sensitive RP[7]. Whereas conformational geometries were investigated on the terminal CS states using time-resolved electron paramagnetic resonance (TREPR)[14,20,22,27], experimental evidence is still lacking on the molecular geometries, the electronic coupling and the electronic energy disorders by protein environment at the secondary CS state $FAD^{-\bullet}\cdots W_B(H)^{+\bullet}$ and at the terminal CS state. One can thus raise the following fundamental questions on the cryptochrome: (1) Are conformation changes of FAD and/or $W_B(H)$ required after the secondary charge separation step to stabilize $FAD^{-\bullet}\cdots W_A(H)\cdots W_B(H)^{+\bullet}\cdots W_C(H)$ with respect to the primary CS state? (2) Which role in magnetic compass sensing does the protein environment including specific water molecules play for the energetics and the electronic character in the CS states?

The TREPR method is particularly powerful for characterizing geometries of photoinduced radical pairs, as recently demonstrated for the primary CS state in plant photosystem II[28], and for the quaternary CS states in cryptochromes from *Chlamydomonas reinhardtii* (*ChlaCry*) and *Drosophila melanogaster* (*DmCry*)[20]. Electron spin polarization (ESP) detected by TREPR as enhanced microwave absorption (A) and/or emission (E) from spin states of spin-correlated radical pairs (SCRP)[14,29–33] is often sensitive to the direction of the external magnetic field ($B_0$) with respect to the molecular frames of the RP constituents (specified by a set of coordinates $\Omega$) because several interactions in the transient CS states are anisotropic. Therefore, one may obtain geometry parameters of transient states by analyzing the ESP pattern of SCRPs. Recently, the electron spin polarization imaging (ESPI) method to map the ESP for several $B_0$-directions was introduced for a clear and direct visualization of geometries of transient CS states[28,34]. From a spectral analysis of TREPR data of the SCRP polarization obtained in thylakoid membranes, the anisotropy of the spin–spin dipolar coupling was mapped to the three-dimensional $B_0$-direction to characterize the positions and the orientation of the primary CS state in the membrane[28]. Geometries and electronic singlet (S)-triplet (T) energy gaps ($2J = E_S - E_T$) were also investigated in CS

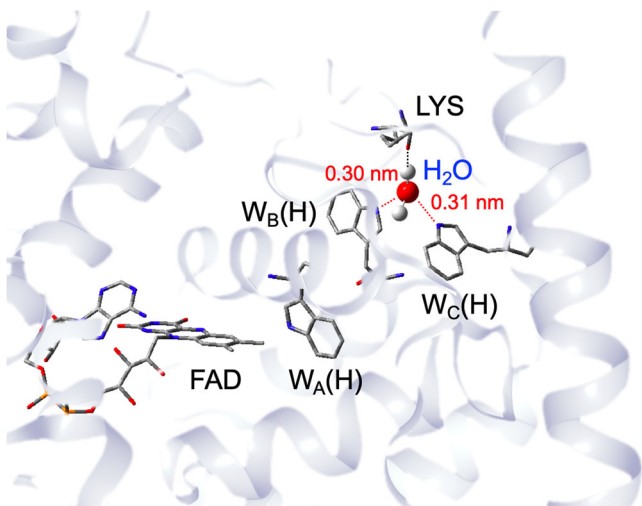

**Fig. 1 Water binding between $W_B(H)$ and $W_C(H)$ forming an electron-tunneling route in chryptochrome.** X-ray structure of the animal-like cryptochrome of *Chlamydomonas reinhardtii* (PDB code:5ZM0) with the Trp triad comprising $W_A(H)$, $W_B(H)$ and $W_C(H)$[17].

states at bulk-heterojunction interfaces of organic solar cells (OCS) by analyses of the SCRP polarization in the donor (D)–acceptor (A) blend thin films of organic semiconductors[34–36]. $2J$ in the CS state has been treated to be a measure of the electronic coupling ($V$) for the long-range ET reaction by the configuration interactions from the charge-recombined singlet ($^1D\cdots^1A$) and triplet ($^3D^*\cdots^1A$ or $^1D\cdots^3A^*$) electronic states through the $V$ interactions[37–39].

The objects of the present study are 1) clarifying molecular geometries of the secondary CS state, i.e. position of oxidized tryptophan and conformation of the reduced FAD in the secondary radical pair, and 2) understanding mechanism of the electron tunneling for generating the terminal CS state in relation to the signaling processes. For this we have measured and analyzed TREPR spectra of wild-type (WT) and mutant cryptochrome DASH from *Xenopus laevis* (*XlCry*-DASH). In the present study, time-dependent energetic disorders in $2J$ are found to be interpreted by a trapping of CS state capturing a reoriented water molecule at 120 K. Enhanced electron tunneling is also revealed by this fluctuating water for the terminal charge-separation at an elevated temperature.

## Results

**Work plan.** We first characterize secondary CS state geometries and the electronic coupling using the light polarization ($L$) effect of the excitation laser on the TREPR spectrum with respect to the direction of the magnetic field ($B_0$) for the WT protein at a cryogenic condition. This $L$ effect, referred to as magnetophotoselection (MPS), was shown to be useful to determine the orientations of spin–spin dipolar interactions[34,40–42]. Secondary, we clarify that energetic disorder in the S–T gap is induced by sub-microsecond water dynamics causing a time-dependent heterogeneity in the exchange coupling of $2J$ in the CS state. Finally, we will discuss in details the electronic coupling matrix elements of the secondary and the ternary CS states by using $2J$ to clarify the role of the water fluctuation dynamics on the charge conduction through the cascade of the tryptophan residues.

**TREPR spectra.** We first observed the TREPR spectra at 240 K (Supplementary Figure 1) of the WT protein to confirm that the reported CS states were obtained as the E/A-polarized TREPR spectra assigned to the terminal RP comprising the reduced FAD and oxidized $W_C(H)$ (=W324 in *XlCry*-DASH)[14,20,27]. Panels a–c of Fig. 2 show TREPR spectra of WT *XlCry*-DASH obtained following photoexcitation with depolarized 450 nm-laser pulses at 120 K. The TREPR spectra are consistent with data reported previously[43] and exhibit E/A/E/A patterns where E and A denote microwave emission and (enhanced) absorption, respectively. From the SCRP model specified in Fig. 2d, the four spin states ($|1\rangle$, $|2\rangle$, $|3\rangle$, and $|4\rangle$) are formed as a result of singlet(S)–triplet($T_0$) interaction in the presence of spin–spin-exchange ($2J$) and dipolar ($d$) coupling, thus leading to four EPR transitions[29,30]. The overall spectral widths of the E/A/E/A patterns depicted in Fig. 2 exceed 10 mT and are considerably larger than those of the RP spectra recorded at higher temperatures. We suggest that spin-spin coupling is larger at 120 K than at 240 K and contributes to spectral broadening, whereas at higher temperatures, the overall spectral width is mainly determined by anisotropic hyperfine interactions in the individual radicals of the SCRP. This implies that the primary or secondary CS state is generated, which leads to stronger spin–spin interactions due to the shorter distances between FAD and $W_A$ or $W_B$ (see Fig. 1) than the distance between FAD and $W_C$.

The overall width of the low-temperature TREPR spectra narrows with increasing delay time $t_d$ after the laser flash, and also the E/A/E/A spectral shape changes, see Fig. 2a–c. This

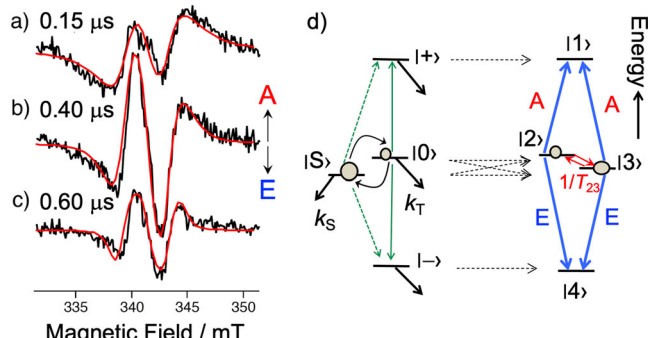

**Fig. 2 Time-resolved EPR (TREPR) of the spin-correlated radical pair. a–c** Delay time ($t_d$) dependence of the TREPR spectrum obtained by the depolarized 450 nm laser irradiation of WT *XlCry*-DASH at 120 K, showing E/A/E/A patterns. **d** The spin correlated radical pair (SCRP) levels ($|1\rangle$, $|2\rangle$, $|3\rangle$, and $|4\rangle$) via the superposition and subsequent decoherences by the interaction between the singlet (S) and $T_0$ ($|0\rangle$) states. The red arrows correspond to population relaxation determined by the rate of $1/T_{23}$ between the $|2\rangle$ and $|3\rangle$ levels contributed by $J$-modulation.

indicates that spin–spin exchange ($2J$) and/or the dipolar ($d$) coupling depend on $t_d$, thus suggesting that submicrosecond dynamic processes modulate the spin–spin interactions in the protein. Previously we presented a matrix formalism based on the Stochastic Liouville Equation (SLE)[35,36,44,45] to analyze TREPR data of transient CS states, which can undergo singlet and triplet charge-recombination kinetics ($k_S$ and $k_T$, respectively in Fig. 2d) and are subject to spin relaxations represented by (i) the relaxation time $T_{23}$ between levels $|2\rangle$ and $|3\rangle$ (two arrows in Fig. 2d right) due to $J$-modulation[7,46], and (ii) the spin-lattice relaxation time $T_1$ determined by fluctuations in the anisotropic g-tensor, hyperfine-tensor and the dipolar coupling[28,47]. These relaxation effects may also alter the TREPR spectral shape and will be discussed below.

**Magnetophotoselection effects on the TREPR spectrum of *XlCry*-DASH.** From the SCRP model, it becomes evident that the two different contributions of $d$ and $2J$ can be distinguished if MPS measurements are employed[34,40]. Fig. 3a shows a view of the principal axis ($d$) of the $d$-coupling between the radicals in the secondary CS state together with the $B_0$ direction, and the transition dipole moment ($M$) for the $S_0$–$S_1$ optical absorption in FAD[48]. The peak-to-peak splitting ($PPS = -4d + 2J$) of the four EPR transitions (A and E in Fig. 2d) is contributed by the spin–spin dipolar coupling represented by $d = D(\cos^2\theta_D - 1/3)/2$ where $D$ represents the dipolar coupling constant. This coupling thus depends on $\theta_D$, i.e. the angle between $B_0$ and $d$ (see Fig. 3a) in the reference X-Y-Z coordinate[49] system, while $2J$ is isotropic[14,34]. This isotopic $J$ is valid in the present radical pairs that exhibit isotropic electronic coupling, as described below.

Figure 3b shows singlet-precursor SCRP spectra for the $B_0$ directions parallel ($B_0 // d$) and perpendicular ($B_0 \perp d$) to the inter-spin vector $d$ in Fig. 3a calculated considering the SCRP level scheme shown in Fig. 2d. The broad E/A polarized spectrum (dashed line) for $B_0 // d$ is due to a large $PPS$ (= 2.4 mT for $\theta_D = 0$), when the Zeeman energy difference ($\Delta g\beta B_0$) determined by the g-factors ($g_{FAD} = 2.0034$ and $g_{W(H)} = 2.0028$)[50,51] of the interacting two radicals is smaller than the $PPS$ obtained for a dipolar coupling constant of $D = -0.90$ mT and an exchange interaction $J = 0.60$ mT. The E/A/E/A spin polarization for $B_0 \perp d$ in Fig. 3b is explained by a small $PPS$ (= 0.6 mT for $\theta_D = \pi/2$) by which the peak splitting is observed at each EPR line for $g_{FAD}$ and $g_{W(H)}$ resulting in two-antiphase doublets. The more intense inner A/E component in the

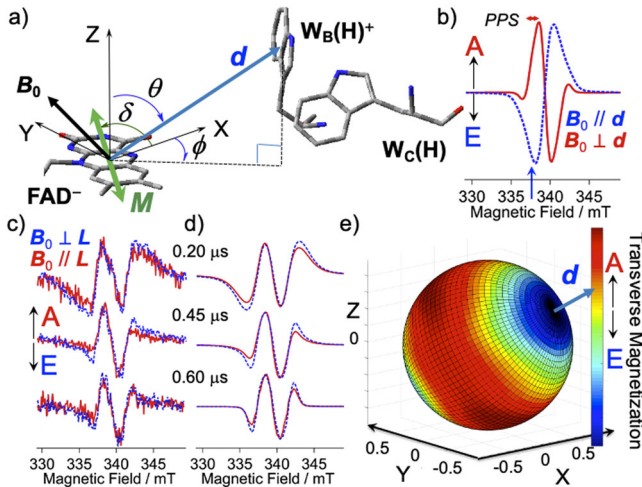

**Fig. 3 Molecular conformation analyses of the secondary RP state. a** Geometry setting of the secondary CS state with the transition dipole moment (**M**) lying in the FAD aromatic X-Y plane with $\delta = 65°$. **b** The singlet precursor SCRP spectra computed for $B_0 \,/\!/\, d$ (dashed line) and for $B_0 \perp d$ (solid line). **c** Magnetophotoselection (MPS) effects of the TREPR spectra for the delay times of $t_d = 0.20$, 0.45 and 0.60 μs at 120 K with $B_0 \perp L$ (dashed line) and $B_0 \,/\!/\, L$ (solid line). **d** Computed EPR spectra of the SCRP for the $B_0 \perp L$ (dashed line) and for $B_0 \,/\!/\, L$ (solid line) with applying $D = -0.90$ mT, $\theta = 58°$, and $\phi = -65°$. $J = 1.45$, 0.55 and 0.40 mT were applied for $t_d = 0.20$, 0.45, and 0.60 μs, respectively. $T_{23} = 0.25$ μs was utilized as the relaxation time constant between |2> and |3> by $J$-modulation, see Fig. 2d. **e** Mapping of the electron spin polarization (ESP) obtained by distributing the transverse magnetization (EPR intensities at $B_0 = 337.50$ mT shown by blue arrow in **b** as the color map to the $B_0$ space directions from the SCRP spectra at $t_d = 0.45$ μs, demonstrating that the **d** vector directs to $W_B(H)$ in the reference X-Y-Z coordinate[49] system in a) of the protein.

red E/A/E/A pattern is due to |2>−|3> population relaxation induced by $J$-modulation, see Fig. 2d[46].

To obtain this **d** anisotropy, we observed the MPS effects shown in Fig. 3c. The outer emissive and absorptive ESP signals of the E/A/E/A patterns are prominent in the spectra (dashed lines) recorded with the polarization of the laser (**L**) perpendicular (⊥) to $B_0$. This indicates that the broad E/A contribution (dashed spectrum in Fig. 3b) is emphasized for $B_0 \perp L$, thus implying that the interspin vector **d** directs far from the direction of **M** because $B_0$ is situated to be perpendicular to **M**. Also, the spectra recorded with $B_0 \,/\!/\, L$ in Fig. 3c are similar in shape to the E/A/E/A pattern (solid line for $B_0 \perp d$) shown in Fig. 3b and are consistent with the above **d** tendency being perpendicular to **M**. These data are consistent with $W_B(H)^{+\bullet}$ being the radical species interacting with $FAD^{\bullet-}$, as shown in Fig. 3a. In Fig. 3d we show simulations of the experimental TREPR spectra using the SCRP model depicted in Fig. 2. The anisotropies of the **g**-matrices and the hyperfine tensors of the individual radicals, $FAD^{-\bullet}$ and $W(H)^{+\bullet}$, together with the spin-relaxation and line broadening parameters compiled in Supplemental Table 1 were used for the spectral simulations[28,34,40]. See also Supplementary Figures 2-8 and Supplementary Tables 2 and 3 for details of the input parameters and their errors. The errors in the angles ($\theta$ and $\phi$) and in the $J$ were evaluated to be ±2 degrees and ±0.1 mT, respectively. The spectra in Fig. 3c were all reproduced with setting the **d**-direction to ($\theta$, $\phi$) = (58°, −65°) with $\delta = 65°$ (Supplementary Figure 2)[48], as shown in Fig. 3d. The data show that at 120 K the secondary CS state is quickly generated within the instrumental response time and that the geometry of the CS state is independent of $t_d$. From

the spectral simulations including the experiments with depolarized light (red lines in Fig. 2)[34], one can distribute the ESP intensities of the transverse magnetization to all the field directions at a specified $B_0$ strength (= 337.5 mT) to obtain the ESPI map shown in Fig. 3e[28]. This visualization clearly shows the orientation of the interspin vector of **d** between $FAD^{-\bullet}$ and $W_B(H)^{+\bullet}$ with respect to the **M** direction in the X-Y reference axes in FAD. The dark red region in the map reflects the strong anisotropy ($A_{ZZ}$) of the nitrogen hyperfine interaction (Supplemental Table 3) in $FAD^{-\bullet}$, further supporting the **d** vector directing from the $FAD^{-\bullet}$ aromatic plane to $W_B(H)^{+\bullet}$. From $D = -0.90$ mT, which reproduced the MPS results in Fig. 3c very well, the separation distance between the spins is estimated to $r_{CC} = 1.45$ (±0.08) nm using the point-dipole approximation. This distance is in good agreement with the center-to-center distance of 1.40 nm between flavin and $W_B(H)$, see Fig. 1. Furthermore, the following conclusions are derived from the MPS analysis and the ESPI mapping:(i) The $FAD^{-\bullet}$ conformation in Fig. 3a remains unchanged with respect to the fully oxidized flavin molecule photoexcited along with the **M** direction in the protein, and (ii) the position of $W_B(H)^{+\bullet}$ is ($\theta$, $\phi$) = (58°, −65°), as in the x-ray crystal structure of Fig. 3a. Because a water molecule is located next to $W_B(H)$ (see Fig. 1), deprotonation could occur to generate the RP $FAD^{-\bullet}\cdots W_B^{\bullet}$ with $H_3O^+$. This possibility is however excluded at 120 K because a large reorganization barrier of 1 eV is expected along the deprotonation reaction coordinate[52]. The spin density distributions predicted by the hyperfine couplings in $W_B(H)^{+\bullet}$ are also coincident with previous report[41,53] as detailed in Supplemental Table 2. We thus conclude that the molecular geometries of FAD and $W_B(H)$ are both preserved after stepwise charge separations at 120 K.

**Modeling time-dependent disorder in the S-T Gap.** In the present line-shape analysis (Supplemental Table 1), heterogeneities in the $3d$ and $d - 2J$ energies are determined by $T_{2d}^*$ and $T_{2J}^*$, respectively[44], and contribute to variation in the $T_0 - T_\pm$ and $S - T_\pm$ gaps (solid and dotted arrows[34,44] in Fig. 2d left), respectively. Thus, $1/(2\pi T_{2d}^*)$ and $1/(2\pi T_{2J}^*)$ are relevant to the $T_0 - T_\pm$ and $S - T_\pm$ variations, respectively, for the EPR transitions in Fig. 2d right. The $t_d$ dependence of the entire spectral line shape in Fig. 2 and in Fig. 3c was explained by a decrease in the $2J$ together with an increase in $T_{2J}^*$ as $t_d$ proceeds, resulting in the narrow E/A/E/A line at 0.6 μs. As a result, ($J$, $T_{2J}^*$) = (1.45 mT, 3 ns), (0.55 mT, 15 ns), and (0.40 mT, 20 ns) were obtained at $t_d = 0.20$, 0.45 and 0.60 μs, respectively, to fit the data as shown in Fig. 3d.

To examine whether the above EPR shape originating from the width represents the heterogeneity of the S–T gap or not, we plotted the EPR line-shape of the antiphase pattern, which is one E/A part of the E/A/E/A antiphase doublets of the SCRP spectrum[29,30]. This is simply confirmed by summing two of the Lorentz functions (Supplementary Equation (1)) in Supplementary Figure 9 assuming $D = 0$ for the |1>−|3> and |2>−|4> transitions. Because PPS of this E/A polarization line (Supplementary Figure 9) is determined by the single input value of $2J$[29,30], and because the spectral width reflects $1/(2\pi T_{2J}^*)$ as an uncertainty in $J$, the high-field side from the center of the E/A line represents a distribution function of the S-T gap. From this, the distribution functions of $2J$ were derived using Supplementary Equation (2) in Supplementary Figure 9, as shown in Fig. 4a for the above ($J$, $T_{2J}^*$) parameters. Notably, the width in the $2J$ distribution was identical with the $1/(\pi T_{2J}^*)$ value (= 4 mT for $T_{2J}^* = 3$ ns) as shown by the solid arrows in Fig. 4a, showing that the $2J$ distributions is dependent of $t_d$. See Supplementary Figures 10–12 for more details on the validity of the present treatment of the $2J$ distributions using the lifetime broadening effect.

From the MPS analysis the geometry of the CS state was revealed to be $t_d$-independent. On the other hand, $2J$ and its distribution both vary with $t_d$. Thus, the decay and the disorder of $2J$ are not likely to originate from geometry changes of the radicals of the CS state but from $t_d$-dependent disorder in the solvent coordinate ($X$) of the CS state, i.e., from solvation dynamics. To ascertain this notion, four potential energy curves ($E_{S1}$, $E_{T1}$, $E_{CS}$, and $E_{S0}$) were drawn for the excited states of the FAD, $^1$FAD*, and $^3$FAD*, the secondary CS state (FAD$^{-}\cdots$W$_B$(H)$^{+\bullet}$), and the ground state (S$_0$), respectively (Fig. 4b). These energies were estimated by using reported excited-state energies[48,54], electrochemical potentials ($E^{red} = -0.38$ V for FAD/FAD$^{-\bullet}$ and $E^{ox} = 1.15$ V for W(H)/W(H)$^{+\bullet}$ vs.

NHE)[52,55] and the reorganization energy of $\lambda = 0.41$ eV applicable to the protein environment[56].

From the configuration interaction model for the long-range CS state systems[37,38], the S-T energy gap of $2J_{CT}$ is induced by electronic-coupling ($V$) perturbations from the CR configurations, as expressed by

$$2J_{CT}(X) = \frac{|V_{S1}|^2}{E_{CS}(X) - E_{S1}(X)} - \frac{|V_{T1}|^2}{E_{CS}(X) - E_{T1}(X)} + \frac{|V_{S0}|^2}{E_{CS}(X) - E_{S0}(X)}$$
(1)

where S1 = $^1$FAD* and T1 = $^3$FAD* in Fig. 4b. From Eq. (1), $2J_{CT}(X)$ was calculated as shown by the bold line in the bottom of Fig. 4b with setting $|V_{S1}| = 6.3$ cm$^{-1}$, $|V_{T1}| = 5.6$ cm$^{-1}$ and $|V_{S0}| = 2.0$ cm$^{-1}$ for the CRs to $^1$FAD*, $^3$FAD* and S$_0$, respectively. These couplings are rationalized by the McConnell superexchange model[57] through W$_A$(H), excluding anisotropy in $J$ as detailed in Supplementary Note 1[58].

$|V_{FAD*,secondary}| = (|V_{S1}| + |V_{T1}|)/2 = 6.0$ cm$^{-1}$ is thus evaluated as the electronic coupling leading to the excited FAD* from the secondary CS state. It is anticipated that the S–T gap significantly decreases as $X$ proceeds, as shown by the red arrow in Fig. 4b because of the increase in the gap between $E_{T1}(X)$ and $E_{CS}(X)$ in Eq. (1). This is relevant for the significant decrease in the peak position of $2J$ (Fig. 4a) by $t_d$. From these peak values of $2J$ (= 2.9 mT, 1.1 mT, and 0.8 mT) in Fig. 4a, one can thus estimate the $X$ positions ($X_p = 0.35$, 0.67 and 0.80, respectively) that give the peak of the $2J$-distribution in $X$, as shown by the vertical dashed arrows for $2J = 2.9$ mT (at $t_d = 0.20$ μs) in Fig. 4b. Accordingly, the distribution function of the CS states are depicted by the normalized gaussian function of $\sqrt{\frac{\lambda}{\pi k_B T}} \exp[-\lambda(X - X_p)^2/(k_B T)]$, as the dotted curves in Fig. 4b. From these distributions and $2J_{CT}(X)$ of the red line in Fig. 4b, one can evaluate the distribution of the $2J_{CT}$ values, as shown in Fig. 4c. The good agreements between Figs. 4a, c demonstrate that the solvation dynamics in Fig. 4b play a role for the time-dependent disorder in $2J$. More details on the connection between the electron transfer mechanism and the present solvation dynamics[59] are described in Supplemental Figure 11.

**Protein solvation dynamics in the CS state.** To evaluate the solvation time constant, the $(1 - X_p)$ values in Fig. 4b were plotted against $t_d$, as shown by the semi-log plot of Fig. 5a. The plot obeys a single-exponential decay (solid line in Fig. 5a) with a lifetime of $\tau = 0.34$ μs in Fig. 4b. Notably, this relaxation time is

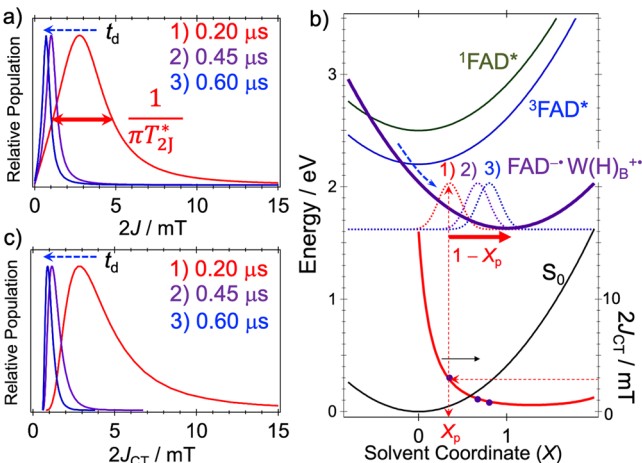

**Fig. 4 Energetic disorder in the spin-spin exchange coupling. a** Time ($t_d$) dependent distributions of the S-T gap ($2J$) of the secondary CS state derived from the higher-field curve (Supplemental Figure 9) of the E/A antiphase EPR line-shape for 1) $t_d = 0.20$, 2) 0.45, and 3) 0.60 μs at 120 K from the right to left, respectively. **b** Potential energy surfaces of the excited singlet state ($^1$FAD*), the triplet state ($^3$FAD*), the secondary CS state (FAD$^{-\bullet}\cdots$W$_B$(H)$^{+\bullet}$) and the ground state (S$_0$). The CS state energy distributions are caused by solvent dynamics, as represented by the Gaussian functions at 1) $t_d = 0.20$, 2) 0.45, and 3) 0.60 μs shown by the dotted lines. The computed S-T gap (= $2J_{CT}$) from the configuration interaction model of Eq. (1) is plotted by the solid red line. **c** The population distributions (dotted lines in b) are plotted as the function of the $2J_{CT}$ (red line in b) for $t_d = 0.20$, 0.45, and 0.60 μs from the right to left, respectively.

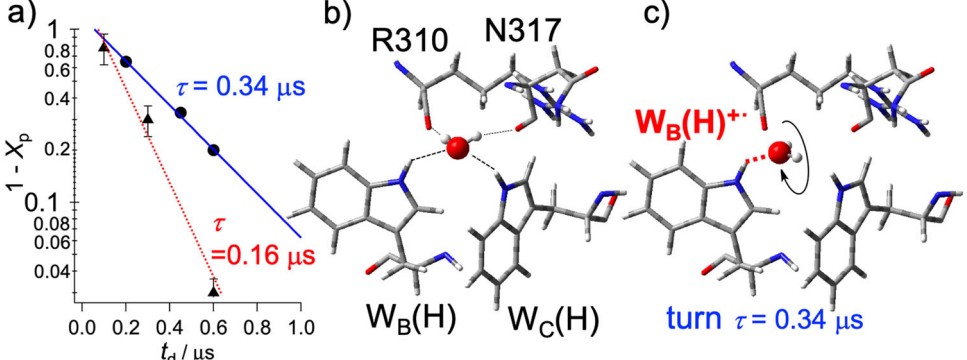

**Fig. 5 Reorientation dynamics of the captured water after the secondary charge-separation. a** Solvation dynamics obtained by the decay of $1 - X_p$ in WT *XlCry*-DASH (●) and in W342F (▲) at 120 K. **b** X-ray structure-based water conformation confined by W$_B$(H), W$_C$(H), and −C=O groups of the backbone at R310 and the N317 residue of *Synechocystis sp.* PCC6803 cryptochrome DASH. **c** Water orientation change by the rehydration (red dotted line) to the cationic charge of W$_B$(H)$^{+\bullet}$ after the secondary CS. The weakened hydrogen bonding between the water and W$_C$(H) hinders the ternary CS by this trapping, indicating the importance of thermal water fluctuations for the terminal CS resulting in W$_C$(H)$^{+\bullet}$ at physiological condition.

coincident with $T_{23} \approx 0.25$ μs utilized as the $J$-modulation induced spin-relaxation[60] between |2> and |3> in Supplemental Table 1. These findings suggest that reorientations of polar groups in amino-acid residues and/or of water molecules nearby FAD$^{-\bullet}$ and $W_B(H)^{+\bullet}$ (W377$^{+\bullet}$ in XlCry-DASH) contribute to relaxation. In particular, a local structural rearrangement close to $W_B(H)^{+\bullet}$ is the most plausible cause after the charge-shift from $W_A(H)^{+\bullet}$ to $W_B(H)$ during stepwise CS. Such slow protein dynamics at cryogenic temperatures were previously reported for the photo-synthetic reaction center from Rps. sulfoviridis and were thought to arise from the reorientation of individual water molecules[56]. Furthermore, one water molecule (Wat830 in Fig. 1) is found between $W_B(H)$ and $W_C(H)$, forming a hydrogen bond network near the protein surface. Hosokawa et al[25]. suggested that the long-range hydrogen bond network is extended by this captured water along the Trp triad and may thus play a role for photo-activation in plant photolyase. This water-binding site (Supplemental Figure 13) was also found in a pigeon cryptochrome of Columbia livia[61].

Figure 5b shows the anticipated molecular conformation of the bound water molecule from the reported x-ray structure of Synechocystis sp. PCC6803 cryptochrome DASH (PDB code: 1NP7)[18]. In addition to $W_B(H)$ and $W_C(H)$, a $-C=O$ group of the backbone at L318 participates in water binding (see Fig. 1), thus resulting in hydrogen bonding of $-C=O\cdots HOH$. The $-C=O$ groups were also identified from R310 and N317 in the cryptochrome DASH of Synechocystis sp. PCC6803: they face toward $W_B(H)$ and $W_C(H)$ (see Fig. 5b) although the captured water molecule was not detected by x-ray crystallography. This indicates that one water molecule is likely to be captured in this specific cavity nearby the protein surface in XlCry-DASH. From the crystal structure shown in Fig. 1[17], the hydrogen bond distance between $W_C(H)$ and the oxygen of Wat830 is $r(N_C\cdots O) = 3.01$ Å; a similar value is found for the respective distance of $W_B(H)$: $r(N_B\cdots O) = 3.00$ Å. This suggests that this bonding network could be utilized for the electron-tunneling routes[62] at the ternary CS to oxidize $W_C(H)$, which will be detailed below.

After the secondary CS at the cryogenic temperature, it is expected that the distance $r(H_B\cdots O)$ becomes shorter on tightening the $H_B\cdots O$ bond due to the positive charge of $W_B(H)^{+\bullet}$, as shown by the red dotted line in Fig. 5c. This rehydration thus weakens the $H_C\cdots O$ interaction to hinder the electronic coupling for the terminal CS, which is consistent with the present assignments as the secondary CS state from the SCRP analysis of the TREPR spectra, see Fig. 3. With reforming the hydrogen bonds, the water molecule is expected to be slowly reoriented by the turn of the O-H group (Fig. 5c). This contributes to the $t_d$-dependent disorder in $2J$ by protein reorganization (see Fig. 4) at $T = 120$ K. The slow solvent dielectric relaxation ($\tau = 0.34$ μs in Figs. 4b and 5c) is not surprising in frozen environments. It is well known that microsecond dielectric relaxations of glass-forming solvents, such as alcohols and 2-methyltetrahydrofuran, determine dynamic spectral shifts and electron transfer (ET) kinetics around 100 K[63-65]. $\lambda = 0.41$ eV, which is much smaller than in bulk water solution[66] (see Fig. 4b), is also well compatible with shifts (ca. 0.4 eV) of the vertical photo-detachment energies in fluorenone anion radicals by forming the 1:1 and 1:2 clusters with methanol in the gas phase[67], thus suggesting that reorientation stabilization associated with a couple of the hydrogen bonds (Fig. 5b) largely contributes to the potential surface in the CS state at this cryogenic condition. This reorganization energy excludes light-induced proton-coupled ET generating FAD$^{-\bullet}\cdots W_B^{\bullet}$ with $H_3O^+$, because a very large reorganization energy of 1 eV is required on the protonation/deprotonation reaction coordinate in Fig. 4b[52].

The dielectric relaxation time ($\tau = 0.34$ μs) is much shorter than the solvation time (>$10^{-6}$ s) in bulk frozen media at 120 K[65]. The present rapid dielectric response thus indicates that the local water molecule participating in the hydrogen bond network[25] is involved in solvation dynamics, as opposed to the condensed phase solvent packing situations in the frozen solution. Slightly larger isotropic proton hyperfine couplings in $W_B(H)^{+\bullet}$ were utilized in Supplemental Table 2 to reproduce the line shapes in Fig. 2 than the couplings for the terminal CS state in Supplemental Figure 1. This might include effects of the superhyperfine couplings[62] of the reorienting water to $W_B(H)^{+\bullet}$ in Fig. 5c.

To examine more details of the solvation dynamics, we observed the TREPR spectra of a mutant of XlCry-DASH, W324F at 120 K, see Supplemental Figure 7a–c. While the E/A/E/A spin polarization pattern of the mutant is very similar to that of the WT, the inner A/E polarization component becomes quickly stronger than the outer E/A component. This is rationalized by the quick |2>−|3> relaxation by the $J$-modulation in Fig. 2d and was consistent with the quick decrease in the $J$ coupling (Supplemental Table 4). From the fitting lines in a)-c) of Supplemental Figure 7 and the time dependence of $2J$, we also estimated $1 - X_P$ values which decay is shown in Fig. 5a to obtain the solvation time of 0.16 μs.

## Discussion

**Water reorientation as the origin of the time-dependent exchange coupling.** The shorter solvation time (0.16 μs from the red dotted line in Fig. 5a) in the mutant as compared to the WT is a strong indication that the local water solvation dynamics dominates the dielectric response and is rationalized by the absence of a hydrogen bond between the water and phenylalanine (see Supplemental Figure 7d). This is because the water molecule is anticipated to rotate more freely in the cavity area, when the hydrophobic phenylalanine residue does not participate in the hydrogen-bond network. In turn, the slower solvation dynamics is considered for the WT to be caused by the reorganization of the hydrogen bond network around $W_B(H)^{+\bullet}$ (Fig. 5b). Notably, the secondary CS state geometries were revealed to be very consistent with the positions of FAD and $W_B(H)$ of the x-ray structures. This is most consistent with the water reorientation mechanism, as detailed below. As an example, if the C=O group of R301 in Fig. 5 directly ligated to $W_B(H)^{+\bullet}$ after the charge-separation at 120 K, the position of $W_B(H)^{+\bullet}$ is required to be changed in the X-Y-Z coordinate system of FAD and must have altered the magnetophotoselection results. In previous MD simulation studies on the ET reactions of cryptochrome and photolyase, conformational changes of the residues nearby $W_B(H)^{+\bullet}$ and FAD$^{-\bullet}$ were predicted during the charge-separation[23]. On the secondary CS state, large degrees of displacements, greater than a few Å, were predicted both for $W_B(H)$ and for threonine residues at room temperature during protein dynamics assisted by the thermal energy for the exergonic $W_B(H)$ oxidation by ligating the polar groups to $W_B(H)^{+\bullet}$. This mechanism is however excluded at cryogenic temperatures, such as the ones considered in this study; thermal activations of protein vibrations were shown to be highly restricted below 150 K[68]. The proposal of water reorientation mechanism with minimal protein displacements (Fig. 5c) is rather reliable scenario for explaining of both (i) the dielectric stabilization dynamics of the secondary CS state and (ii) the blocking of the terminal CS to oxidize $W_C(H)$ at 120 K. The theoretical predictions of the changes in the molecular positions and conformations of the residues and FAD were reasonable when the photoinduced radical species becomes newly bound to one of the polar groups of another residue because the whole

protein molecule possess the self-organized 3D structure via polypeptide chains. On the other hand, the $\boldsymbol{d}$-direction of $(\theta, \phi) = (58°, -65°)$ with $\delta = 65°$ in Fig. 3a is concluded to be time-independent and is consistent with the x-ray structures. This is interpreted by the single water conformation change bound to $W_B(H)^{+\bullet}$ to stabilize the radical pair causing the time-dependent distributions in the S-T gaps as shown in Fig. 4b, although the other environmental effects would participate. FAD is known to be located at the hydrophobic region inside the protein. Thus, the water reorientation around $FAD^{-\bullet}$ is not plausible. Possibilities of glycerol binding effect and its reorientation dynamics are also excluded, as detailed in Supplemental Note 2 with Supplemental Figures 13 and 14.

**Fluctuating captured water promotes electron tunneling in magnetophotoreception.** At 240 K, the SCRP spectra of the terminal CS state, $FAD^{-\bullet}\cdots W_C(H)^{+\bullet}$, were well reproduced (red lines in Supplemental Figure 1) by setting a $t_d$-independent $J$ parameter with $J = 22 \mu T$ for $t_d > 0.1 \mu s$ (see METHODS and Supplemental Table 5) using the x-ray conformations of $FAD\cdots W_C(H)$ shown in Fig. 3a with the inter-spin distance of $r_{CC} = 1.91 \pm 0.08$ nm, as detailed in Supplemental Figure 2. Although this $r_{CC}$ is well compatible with values obtained from EPR studies of the $FAD^{-\bullet}\cdots W_C(H)^{+\bullet}$ distances[14,20–22], the present geometries of the secondary (Fig. 3a) and ternary CS states are in conflict with ultrafast transient absorption anisotropies by which the holes in the sequential CS states were suggested to be delocalized in the Trp-triad[15]. The $t_d$-independent $J$ at 240 K reflects a rapid dielectric response to result in $X_p = 1$ (see Fig. 4b)[69]. Thus, the picosecond vibrational cooling via fast water reorientations induces the localized holes at the sub-microsecond domains, while the CS states could maintain the hole delocalization at the picoseconds time regime[15] because of the involvement of the vibrationally hot CS. Therefore, a quick response via the hot CS in Supplemental Figure 11 would play a role on the secondary CS, as has been discussed on the initial charge-generations at the bulk-heterojunction interfaces of the OCS[26,34,70].

From $X = 1$ in Eq. (1), $|V_{S1,teminal}| = 1.6$ cm$^{-1}$ and $|V_{T1,teminal}| = 1.5$ cm$^{-1}$ were obtained at 240 K as the electronic couplings of the terminal CS state to FAD* using $J = 22 \mu T$ as detailed in Supplemental Table 5. As a result, the electronic coupling ($|V_{FAD*, terminal}| = 1.5$ cm$^{-1}$ to FAD*) in the terminal CS state is attenuated from $|V_{FAD*}| = 6.0$ cm$^{-1}$ of the secondary CS state (120 K). From the superexchange model[57], $|V_{FAD*, terminal}|$ is expressed[45] as follows:

$$\left| V_{FAD^*, \, terminal} \right| = \frac{|V_{HH}||V_{HHAB}||V_{HHBC}|}{\Delta E_{HH}^2} = 1.5 \, \text{cm}^{-1} \quad (2)$$

where $|V_{HH}|$, $|V_{HHAB}|$, and $|V_{HHBC}|$ represent transfer integrals[28,41] between the highest occupied molecular orbitals (HOMOs) as shown in Fig. 6 and in Supplemental Figure 15. $\Delta E_{HH}$ ($\approx \lambda = 0.41$ eV) is the vertical energy gap for the hole-transfer between $W_B(H)$ and $W_C(H)^{+\bullet}$ and is largely caused by the water solvation (Fig. 4b) in the present system. $|V_{HH}| \approx |V_{HHAB}| \approx |V_{HHBC}|$ is assumed[41,45] at higher temperature because corresponding edge-to-edge separations (<0.4 nm in the dashed lines of Fig. 6) are largely common. From Eq. (2), $|V_{HHBC}| \approx 250$ cm$^{-1}$ is thus estimated at 240 K on the hole-transfer between $W_B(H)$ and $W_C(H)$ (see Fig. 6 and Table 1). This value would be too large for the transfer integral between $W_B(H)$ and $W_C(H)$

**Table 1 Electronic Couplings and the Attenuation ($\varepsilon$) from the Secondary to Terminal CS states.**

| CS states (Temperature) | $FAD^{-\bullet}\cdots W_B(H)^{+\bullet}$ (120 K) | $FAD^{-\bullet}\cdots W_C(H)^{+\bullet}$ (240 K) |
|---|---|---|
| $J$ / mT | 0.4[a] | 0.022 |
| $|V_{FAD*}|$ / cm$^{-1}$ | 6.0 | 1.5 |
| $|V_{HHAB}|$ / cm$^{-1}$ | 140 | 250 |
| $|V_{HHBC}|$ / cm$^{-1}$ | – | 250 |
| $\varepsilon$ | 0.08[b] | |
| $\varepsilon_{S1}\varepsilon_{S2}$ | 0.07[c] | |

[a]At $t_d = 0.6 \mu s$ from Fig. 4a. [b] Estimated by $\varepsilon = |V_{HHBC}|/\lambda$. [c] Obtained by the per-unit decays[72] of $\varepsilon_{Si} = (0.36) \times \exp [-11 (R_i - 0.28 \text{ nm})]$ via the hydrogen bonds from Fig. 1.

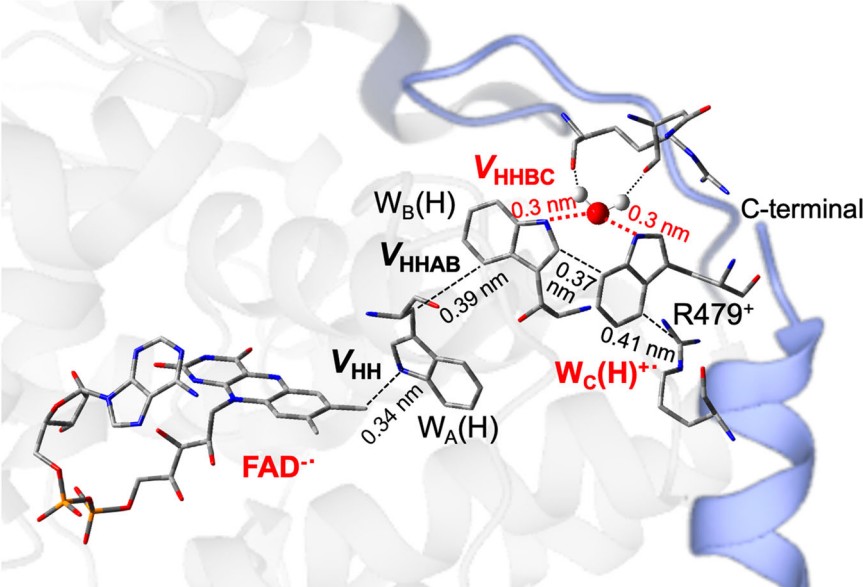

**Fig. 6 Interaction routes in the ternary CS state of $FAD^{-\bullet}\cdots W_C(H)^{+\bullet}$.** Estimated based upon the x-ray structure of *Synechocystis sp.* PCC6803 cryptochrome DASH. Black dashed lines denote nearest-neighbor edge-to-edge separations. $|V_{HH}| \approx |V_{HHAB}| \approx 140$ cm$^{-1}$ was estimated in $FAD^{-\bullet}\cdots W_B(H)^{+\bullet}$ at 120 K (Table 1). Bold dotted lines are the hydrogen bond separations ($R = 0.3$ nm) by the water for mediating the transfer integral ($|V_{HHBC}| \approx 250$ cm$^{-1}$) at 240 K. Structural rearrangements are anticipated in the blue backbones due to the cationic charge generation at $W_C(H)^{+\bullet}$.

because $|V_{HHAB}| = 140\ cm^{-1}$ in the secondary CS state (Supplemental Figure 15) was consistent with the reported couplings ($\approx 100\ cm^{-1}$)[41] for the contact edge-to-edge separations[58]. The electronic coupling was previously demonstrated to be mediated through an intervening bound water in a triplet-triplet energy transfer system at the photoprotective site of the peridinin–chlorophyll a–protein from *Amphidinium carterae*[62]. Thus, $|V_{HHBC}| \approx 250\ cm^{-1}$ is characteristic of the tunneling via the water (see Fig. 6) particularly at higher temperatures.

The attenuation degree ($\varepsilon$) of the electronic coupling by elongation of the distance between the secondary and terminal CS states is represented as, $\varepsilon = |V_{FAD^*,terminal}|/|V_{FAD^*,secondary}|$. $\varepsilon = |V_{HHBC}|/\Delta E_{HH} = |V_{HHBC}|/\lambda$ is derived with substituting $|V_{FAD^*,secondary}| = |V_{HHAB}||V_{HH}|/\Delta E_{HH}$ (Supplemental Figure 16)[45] to Eq. (2), and thus is evaluated to be $\varepsilon = 0.08$ from $|V_{HHBC}| \approx 250\ cm^{-1}$ at 240 K. This perfectly agrees with the *Pathways* model[71] presented by Beratan: the attenuation in $|V_{FAD^*}|$ is $\varepsilon = 0.07$ via the two hydrogen-bond steps (two dotted lines in Fig. 6) as a product ($\varepsilon_{S1}\varepsilon_{S2}$) of the per-unit penalties ($\varepsilon_S$)[72] with $\varepsilon_{Si} = (0.36) \times \exp[-11\ (R_i - 0.28\ nm)]$) by the jumps in the lengths of $R_i = 0.30$ and 0.31 nm between N···O atoms in Fig. 1 (Table 1). Because the *Pathways* model is widely applicable to account for the ET rate constants particularly in systems where tunneling matrix elements are sensitive to the collective thermal equilibrium motions in proteins[72], $|V_{HHBC}| \approx 250\ cm^{-1}$ reflects amplifications of the coupling due to the thermally assisted non-Condon effect in the protein[73]. This is relevant to the entropy mechanism by which the density of states is enhanced by the electron-phonon coupling for the exothermic carrier dissociation in the OCS[26,44,74]. The trapping feature of the secondary CS state is thus relevant to the reported bound electron-hole pairs with separation distances around 2 nm at the D:A interfaces in the OCS, which was explained by the restraint of the electron-phonon coupling at $T = 77\ K$.

The thermally amplified $V_{HHBC}$ in Fig. 6 strongly supports the slow water-trapping scenario on the secondary CS state in Fig. 5c; insufficient water-fluctuations and the resultant rehydration should reduce the $|V_{HHBC}|$ coupling and block the terminal CS generations at 120 K even if the hot CS-state is initially generated[15], leading to the submicrosecond reorientation relaxations. The amplifying phonon modes associated with the bound waters were recently clarified using terahertz spectroscopy on hydrated nylon polymer;[75] the vibration frequencies of the bound-water motions were found to be higher than 4 THz, which corresponds to an energy that is not thermally activated at 120 K with $k_B T = 2.5$ THz but can be accessible at 240 K (5 THz). Kobori et al.[39] reported that the solvent-solvent effective electronic coupling ($|V_{SS}| \approx 850\ cm^{-1}$) was highly amplified by solvent fluctuations at a mean solvent-solvent distance of 0.57 nm in the condensed media from analyses of the chemically induced dynamic electron polarization by radical ion pairs in the liquid solutions. The present enhanced $|V_{HHBC}|$ interaction at 240 K is compatible with this report. Moreover, $|V_{HHBC}|$ which is weaker than $|V_{SS}| \approx 850\ cm^{-1}$ is explained by suppression of the large-scale collective fluctuations at the local protein cavity where significant water-water fluctuations around the protein hydration layer are inhibited.

Finally, a repulsion between $W_C(H)^{+\bullet}$ and $R479^+$ is expected to occur at the ternary CS in Fig. 6 because the arginine residue is deprotonated due to $pK_a = 12.5$. This should cause breaking of the cation-$\pi$ interaction between $R479^+$ and $W_C(H)$ (dashed line in Fig. 6) to release and unfold the C-terminal chain together with subtle rearrangement of the backbone at $R310^+$ and N317 by the rehydration to $W_C(H)^{+\bullet}$. Thereby the captured single water would effectively trigger the light-induced signaling processes at least in *Synechocystis* sp. PCC6803 cryptochrome DASH[76], as well as the regulations in the electron tunneling at the physiological condition, as discussed above.

## Conclusion

We have experimentally characterized the molecular geometries, solvation dynamics and electronic couplings of the secondary and ternary photoinduced charge-separated (CS) states in WT *XlCry*-DASH at $T = 120\ K$ and 240 K, respectively, using the TREPR method. It is concluded that the nuclear displacements by the stepwise charge-separations are minor both in the reduced and oxidized components from the conformations of $(\theta, \phi) = (58°, -65°)$ with $\delta = 65°$ at 120 K. Based upon this, the small degree of the attenuation in the electronic coupling (Table 1) is explained by the motions of the captured single water molecule (see Fig. 6) playing a significant role for mediating the long-range electron-tunneling at 240 K. In particular, thermally assisted water fluctuations at terahertz frequencies[77] are a key factor to prohibit the slow solvation at the secondary CS (Fig. 5c) and to facilitate subsequent charge separations. The present thermal-equilibrium motional assistance coincides with previous predictions by MD simulations[16] that described the picosecond fluctuations in the transfer integral ($V_{HHBC}$) between $W_B(H)^{+\bullet}$ and $W_C(H)$ frequently amplified to $V_{HHBC} > 50\ cm^{-1}$ during the ternary charge-separation event. The solvent-mediated effective tunneling of $|V_{HHBC}| \approx 250\ cm^{-1}$ in the protein cavity is however weaker than the thermally activated inter-solvent coupling ($|V_{SS}| \approx 850\ cm^{-1}$) in the liquid phase. This optimized tunneling matrix element is crucial for regulations of the anisotropic CR of the terminal CS states utilized for the possible magnetic compass senses with the signaling processes[6,8,11], because the terminal RP yield can be determined by the competition[7] between the singlet recombination via $|V_{HHBC}|$ and the anisotropy in the singlet-triplet spin conversion around the strength of the Earth magnetic field.

## Methods

**Sample Preparations**. The gene encoding *XlCry*-DASH, which was cloned into pGEX4T-2 vector[78], was inserted between the NdeI and XhoI sites of a modified vector, pET-28a expression vector (Novagen) whose kanamycin-resistant cassette was replaced with ampicillin-resistant cassette from pET-21a vector (Novagen). A mutant (W324F) of *XlCry*-DASH was constructed by PCR using the QuikChange site-directed mutagenesis method (Agilent Technologies). The WT and mutant *XlCry*-DASH proteins were expressed in E. coli BL21(DE3) as a fusion protein with His$_6$-tag at the N-terminus. Protein expression and purification condition was carried out in accordance with the method of *Xl* (6–4) photolyase[79]. After purification, buffer exchange was carried out by dilution with 0.3 M NaCl, 0.1 M Tris·HCl, pH 8.0, 30% (v/v) glycerol, and concentration by ultracentrifugation devices (Amicon Ultra-15 and -0.5 mL, Merck) for a few times. Because our *XlCry*-DASH contained fully reduced and neutral semiquinoid forms, potassium ferricyanide was added at a final concentration of 5 mM in order to oxidize the samples to fully oxidized form[80]. It took about 5–7 days at 4 °C in the dark to make the samples fully oxidized form. To remove potassium ferricyanide, dilution with the same buffer and concentration by ultracentrifugation was carried out for a few times.

The sample solutions (0.3 M NaCl, 0.1 M Tris·HCl, pH 8.0, 30% (v/v) glycerol) were deoxygenated by the freeze-pump-thaw cycles and were transferred to sample tubes with diameters of 5.0 and 0.6 mm for the measurements at 120 and at 240 K, respectively. These tubes were sealed using a torch for the TREPR measurements. Nitrogen gas was introduced before the sealing of the 0.6 mm-diameter tube.

To check a possible effect of impurity by potassium ferricyanide to oxidize the protein, chromophore-removed *XlCry*-DASH (apo-*XlCry*-DASH) was subjected to ferricyanide treatment. The chromophores were removed by the dialysis against 2 M KBr, 100 mM KCl, 10 mM 2-mercaptoethanol, 1 mM EDTA, and 50 mM Tris-HCl, pH 4.0 at 4 °C for 6 days[81]. The soluble protein was collected and the buffer was exchanged to 0.3 M NaCl, 0.1 M Tris·HCl, pH 8.0, 30% (v/v) glycerol by Amicon devices. The protein concentration was measured using Bio-Rad Protein Assay Kit, based on the Bradford protein assay, with BSA solution as a standard. The treatment and removal of potassium ferricyanide of the apo-*XlCry*-DASH was carried out in the same way as the treatment of the chromophore-bound *XlCry*-DASH.

The molar absorption coefficient of the chromophore removed sample was estimated from the protein concentration, the molecular mass of the calculated His$_6$-tagged *XlCry*-DASH (62.8 kDa), and the absorption spectrum, which was estimated to be $\varepsilon_{257nm} = 3.61 \times 10^5$. UV-vis spectra of apo-*XlCry*-DASH with and without ferricyanide treatment as well as chromophore-bound *XlCry*-DASH were shown in Supplementary Figure 17. Oxidized forms of amino acid species were

evaluated to be minor by the optical absorption bands from 300 to 500 nm for tryptophan[82] and for tyrosine[83]. The positions of the tryptophan and tyrosine side chains that could be oxidized were estimated based on the crystal structure of *Synechocystis sp.* PCC6803 cryptochrome DASH[18] (Supplementary Figure 18).

**Time-resolved EPR measurements.** The X–band TREPR measurements were performed using a Bruker EMX Plus system in which a modified wide-band pre-amplifier was equipped in the microwave bridge. The field modulation was not employed. Light excitations were performed by Continuum optical parametric oscillators (OPO) systems (Surelite OPO Plus) pumped with a third harmonics (355 nm) of a Nd:YAG laser (Continuum, Surelite I-10, 5 ns). A laser de-polarizer (SIGMA KOKI, DEQ 1N) was placed between the laser exit and the microwave cavity for the depolarized TREPR data. For the MPS, a polarizer (SIGMA KOKI, WPQW-VIS-2M) was used to rotate the laser polarization direction by 90 degrees. Transient EPR signals were averaged by a Tektronix DPO3054 500 MHz digital phosphor oscilloscope at 201 different external magnetic field positions[34]. Temperature was controlled by a cryostat system (Oxford, ESR900) by using liquid nitrogen as the cryogen[84].

**Numerical simulations of the TREPR spectra.** Numerical calculations of the EPR spectra were performed using MATLAB (The MathWorks) codes. The computation methods to obtain the delay time ($t_d$) dependence of the TREPR spectrum[28,36,45] and the time profiles[35] of the transverse magnetizations were reported previously for the transient CS states. We set a precursor CS state in which density matrix elements of coherence terms ($\rho_{0S}$ and $\rho_{S0}$) in an S-T$_0$ basis system developed from 0 ns to the time of $t_d$ in the presence of a strong exchange coupling ($J_{pre}$ = 3 mT) for the secondary CS state[34]. Then, such coherence terms together with the populations ($\rho_{SS}$ and $\rho_{00}$) were transferred to the corresponding density matrix elements of the SCRPs to generate the resultant density matrix elements[85]. This will produce an overlapping net 'E' polarization in the FAD$^{-\bullet}$ resonance field and a net 'A' in the higher field W(H)$^{+\bullet}$, which is originating from the chemically induced dynamic electron polarization (CIDEP) from the precursor radical pairs[85–87]. For the MPS analysis[40], the ESP signals ($ESP^0_{para}$ and $ESP^0_{perp}$) for $B_0$ // $L$ and $B_0 \perp L$, were set to be proportional to the squares of direction cosines and to the halves of the squares of the sine components, respectively, between the $B_0$ and $M$ vectors (Fig. 3a). As for the terminal CS state (Supplemental Figure 2) obtained at 240 K, $J_{pre}$ = 0.1 mT was considered. The $\rho_{0S}$ and $\rho_{S0}$ terms were set to be transferred to the corresponding density matrix elements of the SCRPs in a 1 ns time constant because of the quick solvent relaxation in Fig. 4b. In this case, the CIDEP effect was very minor in the ESP.

## Data availability
The data associated with the reported findings are available in the manuscript or the supplementary information. Other related data are available from the corresponding authors upon request.

## Code availability
The MATLAB codes are available from Yasuhiro Kobori upon request.

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

## Acknowledgements
This work was partially supported by JSPS KAKENHI Grant Numbers of 19H00888, 20K21174 and Grant-in-Aid for Transformative Research Areas, "Dynamic Exciton" (JP20H05832) to Y.K. Y.K. appreciate the support by Dr. Hiroki Nagashima (Saitama University) in the TREPR measurements using the cryostat system. YK thanks Professor Kiminori Maeda (Saitama University) for fruitful discussions on the transient intermediate species during the RPs.

## Author contributions
Y.K. and S.W. conceived the project. T.I. and H.K. expressed the proteins and characterized the purifications. M.H., S.W., M.F., and Y.K. performed TREPR measurements. Y.K. developed the theoretical description and the MATLAB codes for analyzing the TREPR data. Y.K., T.I., and S.W. wrote the manuscript with input from all authors.

## Competing interests
The authors declare no competing interests.
