## [Peer Review File · Communications Chemistry]

Reviewers' comments:

Reviewer #1 (Remarks to the Author):

Comments on "Orientations and Water Dynamics of Photoinduced Secondary Charge-Separated States for Magnetoreception by Cryptochrome", by Kobori et al.

The authors are working on an important problem. The unraveling of the role of cryptochromes in animal migration is of great interest to spin chemists, physicists, biologists, and zoologists.

The paper presents time-resolved electron spin resonance data on wild type and mutant cryptochrome systems, and the authors have gone to considerable lengths to investigate the role of a ligating water molecule that creates a trapped charge separated state. The experiments have been carefully planned and executed.

The finding that there is a time dependence to the exchange interaction $2J$ is interesting but not entirely unexpected in a system undergoing a charge transfer cascade.

I am afraid I cannot recommend publication of the paper in its present form. The changes in the TREPR spectra are very subtle and small, and the number of parameters used to fit the data very large, in fact somewhat overwhelming.

I suggest a different approach for the analysis. First and foremost, explain carefully the limitations of the model. Give some idea of the error limits for each parameter, and state which parameters can be determined or at least estimated by other methods so that some independent verification can be established. Once that is done, it would be very helpful to see a series of simulations showing, for example, what happens when $2J$ is varied and D is held constant, and vice versa. This, plus some error bars, would establish significantly higher confidence in the simulations. How many of the parameters need to be adjusted to achieve a unique fit?

A case in point is Figure S3, where the experimental data is of such poor S/N it would be useful to know just how large a variation in simulation parameters (and which ones) will still reproduce the main features of the simulations.

Overall, the paper seems to admit in its language that the interpretation of the results is somewhat speculative: For example in lines 339 and 343 - "may" thus weaken, "may" contribute, and the entire text between lines 465 and 472.

Other points:

$W_b(H)^+$ is not defined in the abstract.

line 109 singlet (S)-minus triplet (T) is awkward.

line 143 why lower case d for dipolar interaction?

line 189-190 - why not use the more commonly understood term zfs (zero-field splitting) instead of

d?

line 191 2J is not always isotropic especially at low temperatures. It is better to say that at a fixed orientation between the unpaired spins and a fixed distance, 2J is expected to be constant.

Reviewer #2 (Remarks to the Author):

What are the major claims of the paper?

The contribution by Weber, Kobori and co-workers describe their efforts in disentangling the intricate relationship between charge-separation states and excited triplet state of the W triad, trying to address underlying questions in charge recombination for magnetoreception in cryptochromes, more in general.

In particular, given their challenging spectroscopic and theoretical analysis, they support a previously proposed hypothesis by which solvation dynamics of a bridging water can control the electronic coupling, thus being the sensitive moiety under physiological conditions. They present TREPR data to support their claims under various conditions of irradiation and temperature.

Though they could not unequivocally demonstrate water-dependent logical gate mechanism, they are still providing a clear and credible description, under relevant conditions, of the mutual orientations of the W triad at least in XICry-DASH. This would exclude some other hypotheses involving Trp-shifts along CT events.

This is a novel insight in the field and of general interest to the wider community working in this field, and therefore, in my opinion, deserves publication in CommsChem.

Nevertheless, I think the authors should address some minor and major issues before publication:

Major 1

I fully understand the authors tried their best in order to tell the story in the most linear way, however I feel that it is very difficult for the reader to follow the flow of the paper without a structured "work plan" paragraph at the beginning of the results and discussion section. The authors provided only a very cryptic and fast spoiler of their results in l. 126-133, but the paragraph is too elusive and does not clearly present the objectives (exception made for too generic questions stated before), and the investigation plan they will adopt to achieve those objectives, in a point-by-point fashion. As an example, I do not feel the sentence in l. 130-131 is at the right place, being probably part of the preamble, thus pausing the presentation of the work plan.

Major 2

Moreover, the last sentence at l. 131-133 is somewhat too strong, given that any direct evidences are given for water binding. I suggest changing to "...is induced by sub-microsecond solvent dynamics under our experimental conditions, causing...".

In this respect, let me raise some very general concerns:

- samples are in 30% glycerol, have the author tried to model glycerol binding in that specific pocket among WB/WC and R310/N317?
- authors performed a long pre-oxidation step to get oxidized FAD, do they have any experimental evidence that any relevant oxidation has involved one of the Trp B and C residues? Even a 20% oxidation, forming oxy-indole moieties would significantly alter the expected results.

Major 3

Do the authors have taken into account to perform some of their experiments with isotopically labeled water? Either ^{18}O or deuterated water, should considerably alter their observation in the acquired spectra, and most probably unequivocally demonstrate a bound-water-dependent mechanism. I am not practical with data acquisition by TREPR technique, so I apologize if I am giving a naive suggestion.

Major 4

Magnetophotoselection effects reported in Figure 3c are very weak, and I am afraid they may not support the theoretical differences as they have been fitted in Fig 3d. In particular, noise in 0.20 microseconds trace is apparently different between red and blue line, and among the red lines in general. The authors should comment on that and eventually provide statistics on data acquisition.

Minor 1

Sentence in l. 149-152 seems to implicate an obvious assumption, whilst it is a consequence of an educated guess, that will be later supported by experiments. I suggest revision.

Minor 2

Italic d in l. 155 has not been declared before.

Minor 3

I would discourage excessive use of abbreviations, especially in the figure captions and conclusion section a part from very common ones. Or at least, repeating again their meaning in figure captions to make them self-explanatory.

Minor 4

In l. 402 the electronic couplings subscript appeared to be "temimal" instead of "terminal"

Curiosity 1

In the conclusions the authors discuss about the real-life conditions. I am not a biologist, so are these hypoxic conditions? There could be any effect of dioxygen in that pocket under physiological conditions? Which would be the implications if that is a pocket for a paramagnetic molecule?

Overall, I think that the work is very much dedicated to a very specific and well-trained audience. I understand that the authors mainly refer to their own community, nevertheless, given that the technique is still not widely adopted and the generalist nature of the journal, they could make a few efforts to make at least introduction and conclusion sections more accessible to a wider audience.

Reviewer #3 (Remarks to the Author):

Hamada et al. here describe a time-resolved electron paramagnetic resonance study of the photoinduced electron transfer in the fully oxidized form of a cryptochrome from *Xenopus laevis*. The authors find that at cryogenic temperatures, the charge separation is halted at the radical pair comprising FAD and WB, the second member of the tryptophan triad. They attribute this to "re-hydration" of the WB radical cation by a captured single water molecule on the microsecond time scale.

In general, this is a very well executed study. Experiments and data analysis have been meticulously conducted. The key finding is the time-dependent exchange coupling at 120 K. The data are then interpreted in terms of captures water dynamics, which in part appears to involve speculative elements and analogy with unrelated systems. While this interpretive part is still executed with skill, I would appreciate if the arguments were laid out in more detail.

For example, the water-reorientation model is introduced by noting that the “findings suggest that reorientations of polar groups in amino-acid residues and/or of water molecules nearby FAD-• and WB(H)+•”, followed by a short mentioning of a photosynthetic reaction center and a photolyase, where this is apparently established. In doing so, the possibility of rearrangements near FAD-• or other structural rearrangements are not discussed at all, despite the fact that “the captured water molecule was not detected by x ray crystallography of *Synechocystis* sp. PCC6803” and lacking direct experimental evidence. While I am still convinced by the chain of arguments constructed by the authors, a broader, more open discussion would be appreciated to stimulate the academic discourse. In particular, it might be a good idea to separate “Results” and “Discussion” (instead of opting for “Results and Discussion”).

The most interesting aspect of the manuscript (apart from the experimental findings) is the interpretation of the time-dependent exchange coupling in terms of slow dynamics along the FAD-WB electron transfer coordinate as summarized in Fig. 4. However, questions remain: Firstly, on the conceptual side, are you suggesting that the radical is actually produced via this ET-coordinate? If so, why is the ET-rate not equally limited by the slow dynamics along this coordinate? If not, why is the water-dynamics strongly coupled to this ET-coordinate, while it (apparently) does not impact the coordinates that give rise to the secondary radical pair (I assume via the primary radical pair)? Further, why does this faster processes take the system to a non-equilibrium configuration of the slow process instead of the equilibrium charge-transferred state given that the dynamics along this coordinate are fast(er)? Could it be that the slow water dynamics are not coupled to the ET-coordinate but rather correspond to a secondary relaxation process following the ET, which modulates relative free energies and/or coupling matrix elements? Why is this option excluded?

Second, on the technical level, I wonder if the primary CT-state should be included in the superexchange model. As the exchange coupling will presumably be huge in this radical pair, it will give rise to different energy shifts in the singlet and triplet configuration of the secondary radical pair. As for the analysis of finding the relevant X_p , it is unclear to me why the configuration should retain the Gaussian distribution corresponding to the equilibrium state (line 296). The dynamics of this processes ought to be diffusive with diffusion coefficient $D = k_B T / \tau_L$, where τ_L is the longitudinal dielectric relaxation time. Again, a Gaussian distribution would be more in line with a model where the water dynamics is not coupled to the ET coordinate, but a subsequent relaxation processes.

I could not grasp how the distribution of the J-values in Fig. 4a was established. Why do the distribution have a strict $J > 0$ imposed? I understand that the model uses an effective relaxation time T_{2J}^* that is then reinterpreted as static heterogeneity, instead of the dynamic relaxation processes that the model implies. Firstly, would a model employing actual static inhomogeneity be preferred? Second, if not, please expand on how the form of the distribution has been actually found (the description in the SI was not too helpful).

Generally, it would be informative to specify the SLE used to model the time-resolved EPR spectra. While I am confident that the 4 relaxation times will probably be found somewhere in the authors' publications, a self-contained exposition is preferred.

Concerning the simulation parameters, what is the meaning of k_T ? It appears to me that the triplet state is not accessible from the radical-pair state (at least not in the equilibrium; otherwise the rate constant should be time dependent in the same way as $J(t)$). If k_T is used to model a spin-independent processes, this would imply $k_S \geq k_T$, which is not the case.

Concerning the statement "the vibration frequencies of the bound-water motions were found to be higher than 4 THz, which corresponds to an energy that is not thermally activated at 120 K but can be accessible at 240 K.", are you thus suggesting that excited vibrational states are responsible in mediating the efficient B-C-coupling? Please elaborate how this is meant? I.e. what characteristic of the vibrationally excited state is relevant?

Eventually, the authors allude to magnetoreception in their conclusions suggesting that the observed phenomenon of charge localisation is critical to the cryptochrome compass. I fail to follow the argument. Do the avian cryptochromes have comparable single-water binding sites? If so, this ought to be stated more clearly. Secondly, how are these findings pertinent to cryptochromes operating at physiological temperatures? I don't think that this connection is strong/relevant and I am puzzled by the fact that the TOC graphic shows a bird instead of a frog.

Figure 2d): the caption does not elaborate on the meaning of the schemes, i.e. left vs. right. Further, the statement that the SCRP levels ($|1\rangle$, $|2\rangle$, $|3\rangle$, and $|4\rangle$) are created by the interaction between the singlet (S) and T0 ($|0\rangle$) states is puzzling as that would involve an increase in Hilbert space dimension.

Overall, this is an excellent piece of work that should be published after consideration of the points raised above.

Title: Orientations and Water Dynamics of Photoinduced Secondary Charge-Separated States for Magnetoreception by Cryptochrome
(Manuscript ID: COMMSCHEM-21-0084-T)

Authors: Misato Hamada, Tatsuya Iwata, Masaaki Fuki, Hideki Kandori, Stefan Weber, and Yasuhiro Kobori

Reviewers' comments:

Reviewer #1 (Remarks to the Author):

Comments on "Orientations and Water Dynamics of Photoinduced Secondary Charge-Separated States for Magnetoreception by Cryptochrome", by Kobori et al.

The authors are working on an important problem. The unraveling of the role of cryptochromes in animal migration is of great interest to spin chemists, physicists, biologists, and zoologists.

The paper presents time-resolved electron spin resonance data on wild type and mutant cryptochrome systems, and the authors have gone to considerable lengths to investigate the role of a ligating water molecule that creates a trapped charge separated state. The experiments have been carefully planned and executed.

The finding that there is a time dependence to the exchange interaction $2J$ is interesting but not entirely unexpected in a system undergoing a charge transfer cascade.

I am afraid I cannot recommend publication of the paper in its present form. The changes in the TREPR spectra are very subtle and small, and the number of parameters used to fit the data very large, in fact somewhat overwhelming.

Author reply 1-1: Only this reviewer would not understand our main conclusion on the mechanism of the electron-transfer cascade rationalized with the water fluctuation at elevated temperatures by the phonon effect. At low temperature, it is expected that the electron-hole pair is trapped at the secondary charge-separated state of $\text{FAD}^- \cdot \text{W}_B(\text{H})^+$ due to the restricted molecular motions. The charge trapping by Coulomb attraction is known in the electron-hole pairs at the bulk-heterojunction interfaces of the photovoltaic cells at low temperatures around the separation distance of < 2 nm and has been rationalized by the lack of the electron-phonon coupling in the polymer domains that play a role for the cascade of the charge conductions at room temperature. When this charge trapping is caused by the slow solvation dynamics which is also well known to occur at cryogenic conditions, this may also result in the charge trapping at the secondary CS state, as originally described in the main text.

Therefore, the following sentence was added at line 475 in the revised manuscript.

“The trapping feature of the secondary CS state is thus relevant to the reported bound electron-hole pairs with separation distances around 2 nm at the D:A interfaces in the OCS, which was explained by the restraint of the electron-phonon coupling at $T = 77$ K.”

I suggest a different approach for the analysis. First and foremost, explain carefully the limitations of the model. Give some idea of the error limits for each parameter, and state which parameters can be determined or at least estimated by other methods so that some independent verification can be established. Once that is done, it would be very helpful to see a series of simulations showing, for example, what happens when $2J$ is varied and D is held constant, and vice versa. This, plus some error bars, would establish significantly higher confidence in the simulations. How many of the parameters need to be adjusted to achieve a unique fit?

Author reply 1-2: We appreciate this valuable comment. In accordance with this comment, we carefully examined susceptibilities of the several parameters on the laser polarization effect of the TREPR spectra. A series of simulations are shown in Figure S4-S8 in the revised version of Supplementary Information. In principle, D and J parameters are distinguished by the magnetophotoselection method because the effect of the anisotropic spin-spin dipolar coupling is sensitive to the laser polarization direction with respect to the magnetic field direction, while the orbital overlap is isotropic for the exchange interaction in the present system. The small but existing differences in the TREPR spectra by the direction of the laser polarization were therefore confirmed by the additional TREPR observations (Experiment 2 in Figure L1), showing that the outer E/A polarization is stronger in the blue spectra ($B_0 \perp L$) than that in the red ones for $B_0 \parallel L$.

Figure L1. Two experimental results of the magnetophotoselection measurements of WT XICRY-DASH at 120 K by the 450 nm laser irradiations.

It is also evident that the entire spectrum widths are significantly larger at 0.2 μ s than the widths at 0.6 μ s. Because the J-coupling is more sensitive to small

displacement between the radicals than the dipolar interaction, one can first assume that the initial broadenings of the TREPR spectra at 0.2 μs originate from the distributions in the exchange coupling rather than in the dipolar coupling constant. Thus $T_{2J}^* = 3$ ns corresponding to $1/(\pi T_{2J}^*) = 4$ mT was applied from the entire spectrum region of 18 mT (top arrows in Figure L1) because the additional broadening effects by the spin-dipolar and the hyperfine couplings exist. In this regard, it is noted that the full of the hyperfine tensors (Table S2 and S3) are invoked from the literatures in the present analysis and are not the fitting parameters. At $t = 0.2$ μs , the fitting parameters are thus, T_{2J}^* , D , J , T_{23} and the angles of θ and ϕ . First, we performed the spectrum computation of the CS state with using $D = -0.9$ mT with $\theta = 58$ degrees and $\phi = -65$ degrees, corresponding to the oxidation of $W_B(\text{H})$ by the photoinduced charge-separation in Fig. 3 at the x-ray geometry. (Figure L2e)

Figure L2. Dependence of the spin-spin dipolar coupling parameters on the computed TREPR spectra for $\mathbf{B}_0 // \mathbf{L}$ (blue) and for $\mathbf{B}_0 \perp \mathbf{L}$ (red). $T_{2J}^* = 3$ ns corresponding to $1/(\pi T_{2J}^*) = 4$ mT was fixed. $T_{23} = 0.32$ μs , $J = 1.45$ mT and $\theta = 58^\circ$ were also applied.

Even though a large magnitude of J was applied ($J = 1.5$ mT), the laser

polarization direction dependence of the TREPR spectra were highly affected by the input parameters on the spin-spin dipolar coupling, as shown in Figure L2. When a weak dipolar coupling constant of $D = -0.4$ mT was applied in a), b) and c) in Figure L2, however, the spectrum differences were too small between $B_0 \perp L$ and $B_0 \parallel L$ at any angle of ϕ and are deviated from the experimental results. For $D = -0.9$ mT (d, e, f in Figure L2) representing the separation distance of 1.45 nm between the radicals, the magnetophotoselection effects become prominent depending upon the position (ϕ) of the radical cation of $W_B(H)$ in the X-Y-Z axis system of the FAD radical anion. The experimental results (light blue and light red lines in Figure L2) were reproduced with $\phi = -65$ degrees (Figure L2e). This is very consistent with the x-ray conformation in Figure 3a, indicating that the $W_B(H)$ is oxidized by the charge-separation at 120 K. When assuming $D = -1.5$ mT from the separation distance of 1.2 nm between FAD and $W_B(H)$ in Figure 1, the experimental results again deviate from the calculations at any position of the oxidized tryptophan (g, h, I in Figure L1).

We also computed the EPR spectra with $D = -5.5$ mT from the separation distance of 0.8 nm between FAD and $W_A(H)$ together with $\theta = 85$ degrees and $\phi = -80$ degrees from Figure 1. The computed spectra were highly deviated from the experimental results, excluding the detection of the primary CS state composed of FAD^- and $W_A(H)^+$. Above results confirm that the TREPR spectra at $0.2 \mu s$ originate from the secondary CS state. From Figure L2d, e, and f, the error in the angles of θ and ϕ are evaluated to be ± 2 degrees. The error in the D coupling was originally evaluated to be ± 0.37 mT in Table S1.

Next, to obtain the validities of the J parameter together with its distribution determined by T_{2J}^* , we examined the susceptibility by the parameters of J and T_{2J}^* , as shown in Figure L3.

Figure L3. Dependence of the exchange coupling parameters (J and T_{2J}^*) on the computed TREPR spectra for $\mathbf{B}_0 // \mathbf{L}$ (blue) and for $\mathbf{B}_0 \perp \mathbf{L}$ (red). $D = -0.90$ mT, $\theta = 58^\circ$, and $\phi = -65^\circ$ were fixed from Figure L2.

From Figure L3, the combination of $(J, T_{2J}^*) = (1.5 \text{ mT}, 3 \text{ ns})$ only reproduced the experimental results when the x-ray conformations of FAD and WB(H) are adopted as the reduced and oxidized species. Because of these strong sensitivities to J and T_{2J}^* , we are confident about the determinations of the several parameters on the molecular geometries and the isotropic exchange couplings from a series of the simulations. For the later delay times at 0.45 μs and 0.6 μs , the similar small but existing differences between the blue and red spectra caused by the laser polarization in Figure 3 (Figure L1) strongly denotes the geometries of the secondary CS states are unchanged by the delay time, while the J and T_{2J}^* are time-dependent, as shown in Figure 4.

Finally, T_{23} value effect was examined (Figure L4). The strong susceptibility of this relaxation parameter was obtained. Furthermore, the calculated spectra obtained by $T_{23} = 0.2$ and $0.5 \mu\text{s}$ are evidently distinguished from the computed results in Figure L2 and L3 using the other parameters. This denotes that the

several input parameters of T_{23} , D , θ , ϕ , J , and T_{2J}^* can be separately determined from the present simulation to simultaneously fit the magnetophotoselection effects for the different delay times. The above arguments were added at Figure S6 in the revised Supplemental Information. Additionally, the following sentence was added in the main text at line 224 in the revised manuscript, as follows.

“The errors in the angles (θ and ϕ) and in the J were evaluated to be ± 2 degrees and ± 1 mT, respectively.”

Figure L4. Dependence of T_{23} value on the TREPR spectrum, showing strong susceptibility of this relaxation parameter.

A case in point is Figure S3, where the experimental data is of such poor S/N it would be useful to know just how large a variation in simulation parameters (and

which ones) will still reproduce the main features of the simulations.

Author reply 1-3: We appreciate this valuable comment. We performed the TREPR measurements with increasing the number of accumulations to improve poor S/N as shown in Figure S7 in the revised Supplementary Information. The quick spin relaxations to result in the stronger inner E/A component than the outer A/E polarization in the E/A/E/A spectrum is evident and is consistent with $T_{23} \approx 0.07 \mu\text{s}$, $T_{2J}^* = 4 \text{ ns}$ and $T_1 \approx 0.2 \mu\text{s}$ in Table S4 described in the revised Supplementary Information. In accordance with the reviewer suggestions, we computed the dependences of the TREPR spectra on the exchange parameters with setting the CS state geometries of Figure S2a (Figure L4). It has been revealed that the line shapes are significantly sensitive to the small variation in the input parameters. Thus, it is concluded that errors in the exchange coupling is evaluated to be $\pm 0.1 \text{ mT}$. This explanation was added at Figure S5 in the revised Supplementary Information.

Figure L4. Dependence of the exchange coupling parameters (J) on the computed TREPR spectra of W324F from XICry-DASH at 120 K.

Overall, the paper seems to admit in its language that the interpretation of the results is somewhat speculative: For example in lines 339 and 343 - “may” thus

weaken, “may” contribute, and the entire text between lines 465 and 472.

Author reply 1-4: Thank you very much. Because of our additional careful analyses of the TREPR data, we are now very confident about the highlighting water dynamics to play a role for the electron tunneling at the different temperatures. We accordingly corrected the expressions of the speculative sentences.

Other points:

$W_B(H)^+$ is not defined in the abstract.

Author reply 1-5: We added “ $W_A(H)$, $W_B(H)$ and $W_C(H)$ ” after “tryptophan residues” in the line 23 of the abstract.

line 109 singlet (S)-minus triplet (T) is awkward.

Author reply 1-6: We corrected to singlet(S)-triplet(T) energy gap. (line 117)

line 143 why lower case d for dipolar interaction?

Author reply 1-7: We treat that the lower case “ d ” to represent the dipolar-dipolar interaction which depends on the angle between the magnetic field and the principal axis of bold “ d ” in Figure 3. In this respect, “ D value” is commonly used to represent the dipolar coupling constant. Thus, we use these expressions.

line 189-190 - why not use the more commonly understood term zfs (zero-field splitting) instead of d ?

Author reply 1-8: We believe that dipolar interaction (d) is more commonly recognized for the broad readers (including NMR investigators and photochemists) not only in the EPR researchers rather than the zfs is.

line 191 $2J$ is not always isotropic especially at low temperatures. It is better to say that at a fixed orientation between the unpaired spins and a fixed distance, $2J$ is expected to be constant.

Author reply 1-9: We appreciate this comment. We added this explanation in the revised version, as follows. (line 199)

“This isotropic J is valid in the absence of ordered paramagnetic surfaces.”⁴⁹

Reviewer #2 (Remarks to the Author):

What are the major claims of the paper?

The contribution by Weber, Kobori and co-workers describe their efforts in disentangling the intricate relationship between charge-separation states and excited triplet state of the W triad, trying to address underlying questions in charge recombination for magnetoreception in cryptochromes, more in general. In particular, given their challenging spectroscopic and theoretical analysis, they support a previously proposed hypothesis by which solvation dynamics of a bridging water can control the electronic coupling, thus being the sensitive moiety under physiological conditions. They present TREPR data to support their claims under various conditions of irradiation and temperature.

Though they could not unequivocally demonstrate water-dependent logical gate mechanism, they are still providing a clear and credible description, under relevant conditions, of the mutual orientations of the W triad at least in XICry-DASH. This would exclude some other hypotheses involving Trp-shifts along CT events.

This is a novel insight in the field and of general interest to the wider community working in this field, and therefore, in my opinion, deserves publication in CommsChem.

Nevertheless, I think the authors should address some minor and major issues before publication:

Author reply 2-1: We are pleased to see that this reviewer recognizes our present novel insight on the role of water mediated electronic coupling.

Major 1

I fully understand the authors tried their best in order to tell the story in the most linear way, however I feel that it is very difficult for the reader to follow the flow of the paper without a structured "work plan" paragraph at the beginning of the results and discussion section. The authors provided only a very cryptic and fast spoiler of their results in l. 126-133, but the paragraph is too elusive and does not clearly present the objectives (exception made for too generic questions stated before), and the investigation plan they will adopt to achieve those objectives, in a point-by-point fashion. As an example, I do not feel the sentence in l. 130-131 is at the right place, being probably part of the preamble, thus

pausing the presentation of the work plan.

Author reply 2-2: We appreciate this valuable comments. We rewrote the Introduction and added the “work plan” paragraph in the Results. (Line 130) The following sentence was added at line 80 in the revised manuscript. “If a reorientation of one water molecule is preferential after the charge-separation, no conformation change in $W_B(H)^+$ is required.”

Major 2

Moreover, the last sentence at l. 131-133 is somewhat too strong, given that any direct evidences are given for water binding. I suggest changing to "...is induced by sub-microsecond solvent dynamics under our experimental conditions, causing...".

In this respect, let me raise some very general concerns:

- samples are in 30% glycerol, have the author tried to model glycerol binding in that specific pocket among WB/WC and R310/N317?

Author reply 2-3: When the glycerol binding is taken instead of the water between $W_B(H)$ and $W_C(H)$, the electron tunneling matrix element must be weakened through increased number of the sigma bonds (3 bonds) from glycerol in the tunneling route via $W_BH \cdots (H)O-CH_3-CH_2(CH_2OH)-O(H) \cdots W_C H$, as an example, while $W_BH \cdots (H)O(H) \cdots W_C H$ do not possess the sigma bonds in the case of the water binding. From the Pathways model with per-unit penalty of $\varepsilon_C = 0.6$ per the aliphatic bond, the attenuation from the secondary CS to the terminal CS is obtained to be $\varepsilon = 0.0025$ instead of 0.07 in Table 1. This is highly deviated from the experimental result of 0.08. Moreover, it is very unlikely that such a large molecule is bound inside the small cavity area of the protein structure while the water is very likely. Because it is widely accepted that glycerol molecules play a role to surround the protein surface area to protect the native protein structure, we exclude the possibility of the glycerol binding. This explanation was added in the Section 5 of Supplementary Information.

Because of our additional careful analyses of the TREPR data, we are now very confident about the highlighting water dynamics to play a role for the electron tunneling at the different temperatures. Please see “Author reply 2-5” for more details below.

- authors performed a long pre-oxidation step to get oxidized FAD, do they have any experimental evidence that any relevant oxidation has involved one of the Trp B and C residues? Even a 20% oxidation, forming oxy-indole moieties would significantly alter the expected results.

Author reply 2-4: The standard reduction potential of potassium ferricyanide is 0.36 V vs. HNE. From the Nernst equation, this value is too low to oxidize even 1 % of the tryptophan residues of proteins with the oxidation potential of 1.15 V described in the main text. As originally shown, we also checked the TREPR at the elevated temperature (Figure S1) and confirmed the photoinduced terminal CS state by the electron-transfer cascade, as reported previously. Because the photo-oxidization of TrpB was only observed at 120 K, the pre-oxidation in one of the Trp B and C residues is reasonably excluded.

Major 3

Do the authors have taken into account to perform some of their experiments with isotopically labeled water? Either ^{18}O or deuterated water, should considerably alter their observation in the acquired spectra, and most probably unequivocally demonstrate a bound-water-dependent mechanism. I am not practical with data acquisition by TREPR technique, so I apologize if I am giving a naive suggestion.

Author reply 2-5: We fully appreciate this valuable suggestion. We are now preparing XI-CRY DASH sample with exchanging the water with D_2O . In particular, it was very difficult to fully exchange with the D_2O buffer solution from the protein sample prepared in the presence of the H_2O buffer. We first tried to oxidize the FAD in the presence of H_2O buffer and concentrate the protein in the D_2O buffer solution. However, the content of the D_2O is revealed to be less than the content of H_2O from the FT-IR measurements. It has been revealed that we need to take more than another month to prepare the D_2O sample. Thus, we did not add the experimental results of the D_2O solution from the following reasons (items 1-5).

Item 1. At the higher temperature of 240 K, the water mediated electron-tunneling in $W_B H \cdots (H) O(H) \cdots W_C H$ is very reasonable from Table 1 and is also self-consistent with the charge-trapping via the solvation dynamics at $W_B(H)$ to break the hydrogen bonding network at the temperature of 120 K that suppresses the protein fluctuations. The present thermal-equilibrium motional assistance well coincides with previous predictions by MD simulations (*J. Am. Chem. Soc.*, 2015, **137**, 1147-1156.) that described the picosecond fluctuations in the transfer integral (V_{HHBC}) between $W_B(H)^{+\bullet}$ and $W_C(H)$ frequently amplified to $V_{HHBC} > 50 \text{ cm}^{-1}$ during the ternary charge-separation event.

Item 2. Importantly, the secondary CS state geometries were revealed to be very consistent with the positions of FAD and $W_B(H)$ of the x-ray structures. This is very important finding in the present study and is most consistent with the water reorientation mechanism: as an example, if the C=O group of R301 in Figure 5 directly ligated to $W_B(H)^{+\bullet}$ instead of the small water molecule after the charge-separation at 120 K, the position of $W_B(H)^{+\bullet}$ should be required to be changed in the X-Y-Z coordinate system of FAD and must have altered the magnetophotoselection results, as was originally stated in the manuscript on the secondary CS state. In particular, the MD simulation studies predicted that the molecular positions and conformations of TRP residues were changed when the photoinduced radical species starts to be bound to one of the polar groups of another residue, which is reasonable because the whole protein molecule possess the self-organized 3D structure via the polypeptide chains. On the other hand, the d -direction of $(\theta, \phi) = (58^\circ, -65^\circ)$ with $\delta = 65^\circ$ in Figure 3a is concluded to be time-independent and is consistent with the x-ray structures as detailed above. This is only interpreted by the small water conformation change bound to $W_B(H)^{+\bullet}$ to stabilize the radical pair, causing the time-dependent distributions in the S-T gaps as shown in Figure 4b. The possibility of water rearrangements near FAD $\rightarrow\bullet$ may be difficult to be excluded. However, FAD is known to be located at the hydrophobic region inside the CRY-DASH protein. Thus, the water reorientation around FAD $\rightarrow\bullet$ is not plausible. This explanation was added at line 416 in the revised manuscript.

Item 3. The conformation change in $W_B(H)^{+\bullet}$ to newly interact with the polar group is excluded at cryogenic temperatures such as the ones considered in this

study; thermal activations of protein vibrations were shown to be highly restricted below 150 K. Our proposal of a water reorientation mechanism with minimal protein displacements (Figure 5c) is rather reliable scenario for explaining of both (i) the dielectric stabilization dynamics of the secondary CS state and (ii) the blocking of the terminal CS to oxidize $W_C(H)$ at 120 K.

Item 4. The above conserved RP conformation with the x-ray structure strongly supports that reorientation of the small water molecule is only one reasonable mechanism to explain all the present several results including the tunneling matrix elements obtained from the spin-spin exchange couplings in Table 1 and the time-dependent widths together with the attenuations in the exchange couplings (Figures 4 and 5).

Item 5. Even if the D2O solution is prepared, it is highly expected that the D2O reorientation motion is not altered when the hydrogen bonding network in Figure 5b is relatively tight, which is very consistent with the present slow reorientation dynamics.

Nevertheless, the D2O effect observation would still be very important experiment to directly demonstrate the role of the one ligating water for the electron tunneling. It might be possible that the vibration frequency by the $-C=O \cdots D_2O$ binding could alter the reorientation motion. This experimental work should thus be forthcoming after careful sample preparations.

Major 4

Magnetophotoselection effects reported in Figure 3c are very weak, and I am afraid they may not support the theoretical differences as they have been fitted in Fig 3d. In particular, noise in 0.20 microseconds trace is apparently different between red and blue line, and among the red lines in general. The authors should comment on that and eventually provide statistics on data acquisition.

Author reply 2-6: As explained at “Author reply 1-2”, the small but existing differences in the TREPR spectra by the direction of the laser polarization were confirmed by the additional TREPR observations (Experiment 2 in Figure L1), showing that the outer E/A polarization is stronger in the blue spectra ($B_0 \perp L$) than that in the red ones for $B_0 // L$. We also carefully examined susceptibilities of

the parameters on the laser polarization effect of the TREPR spectra as shown in Figures L2, L3 and L4. The errors in the parameters are carefully discussed in the revised manuscript. From this we reasonably concluded the time dependent solvation dynamics to influence the singlet-triplet energy gap.

The noises in the red traces are larger than in the blue traces when each spectrum intensity is shown as normalized by the highest intensity position in the field swept data in Figure L1, denoting that the EPR intensity is weaker in the $B_0 \perp L$ case. The weaker $B_0 \perp L$ spectrum than that for $B_0 // L$ is simply explained by an 1/2 factor is multiplied in the intensity at the former case; the spectrum intensities (ESPpara0 and ESPperp0) for $B_0 // L$ and $B_0 \perp L$ are described to be proportional to the squares of direction cosines and to the halves of the squares of the sine components, respectively, between the B_0 and M vectors in the literatures. (*J. Phys. Chem. B* 2000, 104, 17, 4222–4228)

Minor 1

Sentence in l. 149-152 seems to implicate an obvious assumption, whilst it is a consequence of an educated guess, that will be later supported by experiments. I suggest revision.

Author reply 2-7: In accordance with this suggestion, we rewrote the sentence, as, “This implies that the primary or secondary CS state is generated, which leads to stronger spin–spin interactions due to the shorter distances between FAD and W_A or W_B (see Figure 1) as compared to FAD and W_C in the terminal RP state reached at 240 K.” at line 155 of the revised manuscript.

Minor 2

Italic d in l. 155 has not been declared before.

Author reply 2-8: Although this was declared, we declare again for the readability. (line 161)

Minor 3

I would discourage excessive use of abbreviations, especially in the figure captions and conclusion section a part from very common ones. Or at least, repeating again their meaning in figure captions to make them self-explanatory.

Author reply 2-9: In accordance with this comment, we corrected the corresponding usages of abbreviations at Figure captions and at Conclusion.

Minor 4

In l. 402 the electronic couplings subscript appeared to be "temimal" instead of "terminal"

Author reply 2-10: We corrected this typo.

Curiosity 1

In the conclusions the authors discuss about the real-life conditions. I am not a biologist, so are these hypoxic conditions? There could be any effect of dioxygen in that pocket under physiological conditions? Which would be the implications if that is a pocket for a paramagnetic molecule?

Author reply 2-11: The freeze-pump-thaw cycles were performed before the measurements as described in the experimental section. Thus, the effect of the paramagnetic molecule is excluded in the present experiments.

Overall, I think that the work is very much dedicated to a very specific and well-trained audience. I understand that the authors mainly refer to their own community, nevertheless, given that the technique is still not widely adopted and the generalist nature of the journal, they could make a few efforts to make at least introduction and conclusion sections more accessible to a wider audience.

Author reply 2-12: We believe that the revised version should be attractive to a wide audience because we rewrote the introduction and conclusion accordingly. The following finding stated in the conclusion should be accessible to a wider audience (line 513 in the revised manuscript), and thus is worthy of publication in Communications Chemistry.

“It is concluded that while the nuclear displacements by the stepwise charge-separations are minor both in the reduced and oxidized components from the conformations of $(\theta, \phi) = (58^\circ, -65^\circ)$ with $\delta = 65^\circ$, motions of the captured single water molecule (see Figure 6) play a significant role both for the trapping of the secondary CS state at 120 K and for mediating the long-range electron-tunneling at 240 K. In particular, thermally assisted water fluctuations at terahertz frequencies⁷⁵ are a key factor to prohibit the lower-frequency solvation

at the secondary CS (Figure 5c) and to facilitate subsequent charge separations.”

Reviewer #3 (Remarks to the Author):

Hamada et al. here describe a time-resolved electron paramagnetic resonance study of the photoinduced electron transfer in the fully oxidized form of a cryptochrome from *Xenopus laevis*. The authors find that at cryogenic temperatures, the charge separation is halted at the radical pair comprising FAD and WB, the second member of the tryptophan triad. They attribute this to “re-hydration” of the WB radical cation by a captured single water molecule on the microsecond time scale.

In general, this is a very well executed study. Experiments and data analysis have been meticulously conducted. The key finding is the time-dependent exchange coupling at 120 K. The data are then interpreted in terms of captures water dynamics, which in part appears to involve speculative elements and analogy with unrelated systems. While this interpretive part is still executed with skill, I would appreciate if the arguments were laid out in more detail.

For example, the water-reorientation model is introduced by noting that the “findings suggest that reorientations of polar groups in amino-acid residues and/or of water molecules nearby FAD^{-•} and WB(H)^{+•}”, followed by a short mentioning of a photosynthetic reaction center and a photolyase, where this is apparently established. In doing so, the possibility of rearrangements near FAD^{-•} or other structural rearrangements are not discussed at all, despite the fact that “the captured water molecule was not detected by x ray crystallography of *Synechocystis* sp. PCC6803” and lacking direct experimental evidence. While I am still convinced by the chain of arguments constructed by the authors, a broader, more open discussion would be appreciated to stimulate the academic discourse. In particular, it might be a good idea to separate “Results” and “Discussion” (instead of opting for “Results and Discussion”).

Author reply 3-1: We are pleased to see that this reviewer recognizes our

present novel insight on the role of water mediated electronic coupling and the water reorientation response, as the reviewer 2 does. The possibility of rearrangements near FAD \rightarrow • may be difficult to be excluded. However, FAD is known to be located at the hydrophobic region inside the CRY-DASH protein. Thus, the water reorientation around FAD \rightarrow • is not plausible. This explanation was added at line 416 in the revised manuscript.

As was described in the manuscript and in the above "Author reply 2-5", the secondary CS state geometries were revealed to be very consistent with the geometries of FAD and $W_B(H)$ in the x-ray structure. This is very important finding in the present study and is most consistent with the water reorientation mechanism; if a polar group of one residue nearby FAD ligated to FAD \rightarrow • after the charge-separation at 120 K, the position and orientation of FAD \rightarrow • would be required to be changed (as predicted by the previous MD simulations) and must have altered the magnetophotoselection results. The d -direction of $(\theta, \phi) = (58^\circ, -65^\circ)$ with $\delta = 65^\circ$ in Figure 3a is concluded to be time-independent within the errors in the angles of ± 2 degrees as detailed above. This conserved conformation with the x-ray structure strongly supports that the small water molecule's reorientation around $W_B(H)$ is only one reasonable mechanism to explain all the present several results including the tunneling matrix elements obtained from the spin-spin exchange couplings in Table 1 and the time-dependent widths together with the attenuation in the exchange couplings (Figures 4 and 5). This explanation was added at line 394 in the revised manuscript.

As suggested, we separated "Results" and "Discussion" in the revised manuscript.

The most interesting aspect of the manuscript (apart from the experimental findings) is the interpretation of the time-dependent exchange coupling in terms of slow dynamics along the FAD-WB electron transfer coordinate as summarized in Fig. 4. However, questions remain:

Firstly, on the conceptual side, are you suggesting that the radical is actually produced via this ET-coordinate? If so, why is the ET-rate not equally limited by the slow dynamics along this coordinate? If not, why is the water-dynamics

strongly coupled to this ET-coordinate, while it (apparently) does not impact the coordinates that give rise to the secondary radical pair (I assume via the primary radical pair)? Further, why does this faster processes take the system to a non-equilibrium configuration of the slow process instead of the equilibrium charge-transferred state given that the dynamics along this coordinate are fast(er)? Could it be that the slow water dynamics are not coupled to the ET-coordinate but rather correspond to a secondary relaxation process following the ET, which modulates relative free energies and/or coupling matrix elements? Why is this option excluded?

Author reply 3-2: We appreciate this valuable comment. In the ET reaction, several nuclear coordinates are considered to be participating including the energy relaxation process. The response time may vary with nature of the coordinate. This concept was originally proposed by Sumi and Marcus (the Sumi-Marcus model: Figure L5). In the present case, the secondary CS state, i.e. $\text{FAD}^{\bullet-} \text{WB(H)}^{\bullet+}$ could be generated at picoseconds regime via the vibrationally hot exciton in the primary CS state of $\text{FAD}^{\bullet-} \text{WA(H)}^{\bullet+}$. In this regard, the very quick nuclear and/or solvent motions may be involved forming the secondary CS immediately as reported in *J. Am. Chem. Soc.*, 2019, 141, 13394-13409. Thus, as the reviewer pointed, the slow water dynamics is regarded as the secondary relaxation. In the present model of Figure 4b, however, the reorganization energy was assumed to be determined by the single λ value for the simplicity of the treatment to predict the heterogeneous distribution of the exchange coupling (Figure 4c). We added this explanation at Figure S8 in the Supplementary Information.

Figure L5. Sumi-Marcus model representing the fast nuclear response via vibration motions (vertical axis) and the slow response by the water reorientation (horizontal axis which is X in Figure 4b) in the present study. The sub-microsecond response in Figure 5a is regarded as the secondary relaxation in the product state by the slow reorientation at 120 K.

Second, on the technical level, I wonder if the primary CT-state should be included in the superexchange model. As the exchange coupling will presumably be huge in this radical pair, it will give rise to different energy shifts in the singlet and triplet configuration of the secondary radical pair. As for the analysis of finding the relevant X_p , it is unclear to me why the configuration should retain the Gaussian distribution corresponding to the equilibrium state (line 296). The dynamics of this processes ought to be diffusive with diffusion coefficient $D = kBT/\tau_L$, where τ_L is the longitudinal dielectric relaxation time. Again, a Gaussian distribution would be more in line with a model where the water dynamics is not coupled to the ET coordinate, but a subsequent relaxation processes.

Author reply 3-3: We take that this is an important question regarding analysis of Figure 4. As this reviewer points, we already took such different singlet and triplet energy shifts using eq.(1) caused by the configuration interactions and

thus obtained the J-coupling (red line in Figure 4b) with its distribution (Figure 4c) of the secondary CS state. The primary CS character can be participating in the secondary CS state as the wavefunction admixture via the electronic coupling. From the perturbation theory, this coefficient of the wavefunction participating to the secondary CS is readily evaluated to be $V_{\text{HHAB}}/\lambda \approx (140 \text{ cm}^{-1}/3200 \text{ cm}^{-1}) = 0.04$ from Figure S13 that explains the time-dependence of the S-T gaps with their distributions (Figure 4a). This denotes only 0.2 % of the primary CS character via the superexchange model around $X = 1$ in Figure 4b, meaning that the primary CS character is much smaller than 1 % even at 0.2 microsecond because the solvation relaxation already proceeded with $X_p = 0.4$ in Figure 4b. This well coincides with the magnetophotoselection results that showed the time-independent α -direction of $(\theta, \phi) = (58^\circ, -65^\circ)$ with $\delta = 65^\circ$ in Figure 3a, as the dominant CS state. This description was added at Figure S14 in the Supplemental Information.

We approximated the distribution by the Gaussian functions to simply evaluate the inhomogeneous J-distributions, although the non-relaxed states were treated. This is because a previous study apparently exhibited the Gaussian distribution shapes to explain the solvation dynamics (*J. Phys. Chem.* 1991, 95, 25, 10475–10485). The present simplified assumption by the equilibrium Gaussian distribution is probably the reason for the slight differences between the distributions around the low-J regions in Figure 4a and 4c because the distribution width is anticipated to be larger than the standard deviation of the equilibrium Gaussian distributions. Although more rigorous treatments invoking the Langevin equation should be forthcoming, which is out of scope in the present interpretations, there is no doubt about the present main conclusion on the involvements of the slow water reorientation causing the J-distribution.

I could not grasp how the distribution of the J-values in Fig. 4a was established. Why do the distribution have a strict $J > 0$ imposed? I understand that the model uses an effective relaxation time T_{2J^*} that is then reinterpreted as static heterogeneity, instead of the dynamic relaxation processes that the model implies. Firstly, would a model employing actual static inhomogeneity be

preferred? Second, if not, please expand on how the form of the distribution has been actually found (the description in the SI was not too helpful).

Author reply 3-4: The positive J is mainly from the second term of eq.(1) which contribution is dominant because $|E_{CS}(X) - E_{T_1}(X)|$ is smaller than $|E_{CS}(X) - E_{S_1}(X)|$ and $|E_{CS}(X) - E_{S_0}(X)|$. Because the CS state energy is lower than the excited triplet energy and is also near to the triplet state at any solvent coordinate in the present system, J needs to be positive at any X .

We examined a microwave power effect of the TREPR signal at 120 K and observed a spin nutation with $B_1 \sim \text{c.a. } 10^6 \text{ rad/s}$, as shown in Figure L6. This wavy profile excludes the dynamic T2 relaxation with the nanosecond time regime and is consistent with a larger longitudinal spin relaxation times in Table S1, denoting that the line shapes are determined by the inhomogeneous distribution of the exchange coupling. This explanation was added in Figure S12 in the Supplemental Information.

Figure L6. Time profiles of the transverse magnetization of WT *XlCry*-DASH at 120 K with two different microwave powers (1 mW and 3 mW for the red and blue profiles at $B_0 = 340 \text{ mT}$). Rabi oscillation frequency is 1.7 time higher in the blue profile than that in the red profiles and thus is caused by the transient nutation, demonstrating that the T_1 and T_2 relaxations are larger than 1 microsecond.

Figure L7. a) SCRPs spectra (transverse magnetization plotted as a function of B_0) computed with different J coupling values with $T_{2J}^* = 15$ ns with the depolarized laser excitations. Probabilities (p) of the exchange couplings were determined from the distribution function (Figure 4a) in b). c): (Red) Averaged SCRPs spectrum summed with weighting the p factors from the transverse magnetizations in a). (Blue) The SCRPs spectrum obtained using $T_{2J}^* = 4$ ns and $J = 1.45$ mT.

To further confirm the validities of the static distribution effects represented by the single Lorentzian width of $T_{2J}^* = 4$ ns (Figure S10), we computed the transverse magnetizations (i. e. the spin-polarized SCRPs spectra) for the several different J parameters, as shown in Figure L7a with setting $T_{2J}^* = 15$ ns. For the different J values, we also obtained probabilities (p) of the J couplings in the protein environment from the distribution function considered by eq.(S2). The averaged EPR spectrum with weighting the p factor was calculated as shown by the red line in Figure L7c. This was very consistent with the spectrum (blue line) computed with the single parameter of $J = 1.45$ mT with $T_{2J}^* = 4$ ns, denoting a very good compatibility representing the J -distribution by the present

computation method originally reported in our previous studies. (*J. Am. Chem. Soc.*, 2016, **138**, 5879-5885. and *J. Phys. Chem. Lett.*, 2017, **8**, 1179-1184.)

Generally, it would be informative to specify the SLE used to model the time-resolved EPR spectra. While I am confident that the 4 relaxation times will probably be found somewhere in the authors' publications, a self-contained exposition is preferred.

Author reply 3-5: Based upon our previous report (*J. Phys. Chem. Lett.*, 2017, **8**, 1179-1184.), the following relation is obtained on the microwave transition (ρ_{S+}) affected by the exchange coupling as expressed by the stochastic-Liouville equation:

$$\begin{pmatrix} \rho_{S+} \\ \rho_{0+} \end{pmatrix} = \omega_1 \begin{pmatrix} -Q_+ - d + 2J + \omega_0 - i \left(\frac{k_S + k_T}{2} + \frac{1}{T_{2J}^*} \right) & Q_- \\ Q_- & -Q_+ - 3d + \omega_0 - i \left(k_T + \frac{1}{T_{2d}^*} \right) \end{pmatrix}^{-1} \begin{pmatrix} \rho_{S0} \\ \rho_{00} - \rho_{++} \end{pmatrix}$$

where ρ_{S0} denotes the S-T₀ coherence developed by frequency of the energy difference between the |2> and |3> levels. Q_+ and Q_- are determined by the sum and difference in the Larmor frequencies of the two radicals in the SCRPs, respectively. When the Q_- term is ignored as $2|J| \gg Q_-$, the S-T₊ transition spectrum is described by the imaginary part of the ρ_{S+} under the very weak microwave strength (ω_1). This spectrum corresponds to the first term of eq.(S2) when d , k_S and k_T terms are ignored. T_{2J}^* thus causes a lifetime broadening as the uncertainty in the $2J$ value determined by the width $1/(\pi T_{2J}^*)$ as shown in Figure 4a in the main text. Notably, the SCRPs spectra were computed using eq.(S3) without ignoring the above ignored parameters to reproduce the experimental results. This explanation was added in Figure S9 of the Supplementary Information.

Concerning the simulation parameters, what is the meaning of k_T ? It appears to me that the triplet state is not accessible from the radical-pair state (at least not in the equilibrium; otherwise the rate constant should be time dependent in

the same way as $J(t)$). If k_T is used to model a spin-independent processes, this would imply $k_S \geq k_T$, which is not the case.

Author reply 3-6: We apologize that we wrote wrong k_T values ($5.0 \times 10^5 \text{ s}^{-1}$) in Table S1 and S4 by typo. We performed the calculation using $k_T = 5.0 \times 10^4 \text{ (s}^{-1}\text{)}$ instead. We truly thank the reviewer for pointing this. Because the k_T values ($< 10^5 \text{ s}^{-1}$) are too small to affect the EPR spectra at the present time range of $< 1 \mu\text{s}$ and because the triplet state is not accessible, this k_T process does not affect the present conclusion at all. Thus, the triplet recombination kinetics is negligible in the present study.

Concerning the statement “the vibration frequencies of the bound-water motions were found to be higher than 4 THz, which corresponds to an energy that is not thermally activated at 120 K but can be accessible at 240 K.”, are you thus suggesting that excited vibrational states are responsible in mediating the efficient B-C-coupling? Please elaborate how this is meant? I.e. what characteristic of the vibrationally excited state is relevant?

Author reply 3-7: Yes. Because the thermal energy at 240 K is $k_B T = 5 \text{ THz}$ and is larger than 4 THz, the 4 THz phonon mode is vibrationally excited to activate the thermal fluctuation of the electron tunneling between the WB and WC. On the other hand, $k_B T = 2.5 \text{ THz}$ at 120 K denotes that such a thermal fluctuation of the bound water molecule is not vibrationally activated at the lower temperature. This well correlates with the present trapping of the secondary CS state by the slow water reorientation, as was detailed in our original manuscript. We thus added “with $k_B T = 2.5 \text{ THz}$ ” after “120 K” at line 494. Details of the bound water motions are not clear in the protein and are out of the scope. However, it is anticipated that such motions are like the mode by collective motions including O-H stretching found at 5.61 THz region in hydrated nylon polymer. Details are reported in *J. Phys. Chem. B*, 2020, **124**, 422-429.

Eventually, the authors allude to magnetoreception in their conclusions suggesting that the observed phenomenon of charge localisation is critical to the cryptochrome compass. I fail to follow the argument. Do the avian cryptochromes have comparable single-water binding sites? If so, this ought to

be stated more clearly. Secondly, how are these findings pertinent to cryptochromes operating at physiological temperatures? I don't think that this connection is strong/relevant and I am puzzled by the fact that the TOC graphic shows a bird instead of a frog.

Author reply 3-8: It is not clear whether the avian cryptochromes have comparable single-water binding sites. However, it is highly anticipated that there exists such a binding site for a wide variety of the cryptochrome family that play roles on the signaling processes via the electron transfer cascade of the TRP residues because of homology in three-dimensional fold, conservation of critical amino acids, as described in the manuscript. Considering that the basic principle of generating the weak-field sensitive radical pair is still poorly understood on the signaling processes not only for the avian magnetic compass but also for the other processes in plants and animals, it must be a reasonable idea and not puzzling that we adopt one representative example of the magnetic field sensing of birds in the TOC graphic, even though *XiCry*-DASH was used to physically elucidate the general mechanistic model of the electron tunneling occurring only at the physiological temperature. It is also emphasized that the present novel finding of the above water mediation mechanism is now unveiled by the highlighting results of the low-temperature measurements (See Table 1). Not only this water fluctuation for generating the terminal RP, the regulated tunneling by V controlled by the water can make the final RP yield anisotropic, because anisotropic singlet-triplet conversions can be competitive with the singlet-recombination in the radical pairs occurring through the V term at the physiological temperature.

Figure 2d): the caption does not elaborate on the meaning of the schemes, i.e. left vs. right. Further, the statement that the SCRPs levels ($|1\rangle$, $|2\rangle$, $|3\rangle$, and $|4\rangle$) are created by the interaction between the singlet (S) and T0 ($|0\rangle$) states is puzzling as that would involve an increase in Hilbert space dimension.

Author reply 3-9: We rewrote the corresponding figure caption as follows. (line 173 in the revised manuscript)

“d) The spin correlated radical pair (SCRPs) levels ($|1\rangle$, $|2\rangle$, $|3\rangle$, and $|4\rangle$) via the superposition and subsequent decoherences by the interaction between the

singlet (S) and T_0 ($|0\rangle$) states.”

Overall, this is an excellent piece of work that should be published after consideration of the points raised above.

We strongly feel that this revised version is worthy of publication in *Communications Chemistry*.

REVIEWERS' COMMENTS:

Reviewer #1 (Remarks to the Author):

The authors have made considerable effort to improve this manuscript. As I indicated in my first review, this is an important and difficult research problem. The results are interesting and I greatly appreciate the effort here to broaden the text for a more general audience.

While I still think the authors have still somewhat sidestepped my question about the orientation dependence of J in frozen systems, this is an ongoing problem in the field of spin chemistry that deserves additional inquiry, so for now I will let the matter rest.

Publication is recommended.

Reviewer #2 (Remarks to the Author):

I mostly appreciated the author's replies, exception made to Major 2 and 3.

Major 2

Even though the authors provide a thorough analysis of the fitting parameters, I still cannot see any experimental evidence of the bound water under their experimental conditions. The TREPR data is actually following unpaired electrons, and only indirectly implying water role.

I want to be very clear: the authors convinced me that their explanation is a good point in explaining their data, and I believe that water has a big role. However, I am afraid that in absence of a direct evidence, the authors should not come to a definitive conclusion. Nevertheless, if the authors still want to make their assertions, they obviously have to, but they should write in the sentence that this is their interpretation of the data.

About glycerol:

While I may agree on the arguments about glycerol reorientation, I invite the authors to give a look at the Protein Data Bank and see how many times glycerol is observed in place of water in the interior of proteins in place of water pockets or substrates. So the sentence "Moreover, it is very unlikely that such a large molecule is bound inside the small cavity area of the protein structure while the water is very likely." should be removed, unless the authors perform at least a computational docking experiment to prove their statement.

About Trp inadvertent oxidation:

I agree about the Nernst equation, but probably the authors are forgetting that (1) there is a concentration factor in the Nerst equation; (2) Trp oxidation potential may significantly vary because of protein surrounding, hydrogen bonding, dielectric constant, and pH. Please see reference: <https://www.sciencedirect.com/science/article/pii/S0927776511002670> Moreover, even a very small amount of free iron impurity (coming from ferricyanide) may represent a significant amount of catalyst in imidazole monooxygenation (as can be seen in the literature). Why the authors do not provide a simple UV spectrum of the protein without any bound FAD that has been subjected to the same procedure ? This would immediately inform about the Trp(O) formation.

Major 3

I thank the authors for taking into account my suggestion. I am very sorry they were not able to prepare the deuterium exchanged sample. It would have given a lot of information, given that they propose a phonon mechanism of coupling. I was not able to fully understand the problem of D2O exchange they are facing, but I hope they could overcome these problems and publish their results. I think that an isotopic effect would be a clear evidence of the water tumbling effect.

Overall I would accept the manuscript for publication after those minor revisions.

Reviewer #3 (Remarks to the Author):

The authors have updated the manuscript to meet my expectations. All raised issues have been dealt with. While the manuscript could be published in its current form, I have one final suggestions: the arguments now presented with Figure S11 could/should be added to the main manuscript, because they are essential to the conceptual underpinning of the model suggested by the authors.

Title: Orientations and Water Dynamics of Photoinduced Secondary Charge-Separated States for Magnetoreception by Cryptochrome
(Manuscript ID: COMMSCHEM-21-0084-T)

Authors: Misato Hamada, Tatsuya Iwata, Masaaki Fuki, Hideki Kandori, Stefan Weber, and Yasuhiro Kobori

REVIEWERS' COMMENTS:

Reviewer #1 (Remarks to the Author):

The authors have made considerable effort to improve this manuscript. As I indicated in my first review, this is an important and difficult research problem. The results are interesting and I greatly appreciate the effort here to broaden the text for a more general audience.

While I still think the authors have still somewhat sidestepped my question about the orientation dependence of J in frozen systems, this is an ongoing problem in the field of spin chemistry that deserves additional inquiry, so for now I will let the matter rest.

Publication is recommended.

Author reply 1-1: We appreciate this valuable comment. We are pleased to see that this reviewer recognizes our present novel insight on the role of water mediated electronic coupling. It is reported that a perturbation from the excited state may allow the spin-orbit coupling (SOC) to participate in the exchange coupling, resulting in the anisotropy in the exchange parameter, as the origin of the g-factor is in organic radical species in frozen systems. (Bencini and Gatteschi, "EPR of Exchange Coupled Spins" (2012) Dover Publication Inc. pp. 27.) Based upon their formalism, this anisotropic exchange interaction is negligibly minor in the present CS state because 1) third order perturbation treatment of the SOC via the excited state is required and 2) the excited state

energy in FAD-* possessing the n-orbital character must be very high, which is relevant to very small g-anisotropy in Supplementary Figure 2.

Furthermore, from our present analyses of the electron spin polarization, it was demonstrated that the time-dependent J is caused by isotropic transfer integrals of V between the separated $\text{FAD}^{\cdot-}$ and $\text{W}_B(\text{H})^{\cdot+}$ radicals with eq.(1). As described in the manuscript, the transfer integrals are simply determined by the orbital overlap of V_{HH} between FAD and $\text{W}_A(\text{H})$ and by V_{HHAB} between $\text{W}_B(\text{H})$ and $\text{W}_A(\text{H})$ via the bridge-mediated tunneling interactions (Supplementary Figure 13). These orbital overlaps are all isotropic interactions with $|V_{\text{HHAB}}| = 140 \text{ cm}^{-1}$. Because this coupling term is well consistent with the transfer integral at the contact edge-to-edge separation (0.39 nm) between $\text{W}_A(\text{H})$ and $\text{W}_B(\text{H})$, as reported by I. A. Solov'yov (DOI: 10.1038/srep18446), it is concluded that the S-T gaps in Fig.4 are all isotropic in the present system. This means that the J-coupling in the present distant radical pair is not anisotropic and is dominated by the isotropic transfer integrals as the configuration interaction. This is self-consistent with the long-range electronic coupling causing the distant exchange interaction isotopically at 1.4 nm for the secondary CS state. Overall, the anisotropic exchange coupling is concluded to be negligible in the present study. We added this discussion in Supplementary Note 1 in the revised manuscript.

Reviewer #2 (Remarks to the Author):

I mostly appreciated the author's replies, exception made to Major 2 and 3.

Major 2

Even though the authors provide a thorough analysis of the fitting parameters, I still cannot see any experimental evidence of the bound water under their experimental conditions. The TREPR data is actually following unpaired electrons, and only indirectly implying water role. I want to be very clear: the authors convinced me that their explanation is a good point in explaining their data, and I believe that water has a

big role. However, I am afraid that in absence of a direct evidence, the authors should not come to a definitive conclusion. Nevertheless, if the authors still want to make their assertions, they obviously have to, but they should write in the sentence that this is their interpretation of the data.

Author reply 2-1: We appreciate this valuable comment. In accordance with this suggestion, we corrected the corresponding sentences, as follows.

At abstract (line 31): “We found a time-dependent energetic disorder in $2J$ and was interpreted by a trap secondary CS state capturing reorientated water molecule at 120 K.”

At line 376, “This is interpreted by the single water conformation change bound to $W_B(H)^+$ to stabilize the radical pair causing the time-dependent distributions in the S-T gaps as shown in Figure 4b, although the other environmental effects would participate.”

At the conclusion (line 461) , we rewrote the sentences, as follows:

“It is concluded that the nuclear displacements by the stepwise charge-separations are minor both in the reduced and oxidized components from the conformations of $(\theta, \phi) = (58^\circ, -65^\circ)$ with $\delta = 65^\circ$ at 120K. Based upon this, the small degree of the attenuation in the electronic coupling (Table 1) is explained by the motions of the captured single water molecule (see Figure 6) playing a significant role for mediating the long-range electron-tunneling at 240 K.”

About glycerol:

While I may agree on the arguments about glycerol reorientation, I invite the authors to give a look at the Protein Data Bank and see how many times glycerol is observed in place of water in the interior of proteins in place of water pockets or substrates. So the sentence “Moreover, it is very unlikely that such a large molecule is bound inside the small cavity area of the protein structure while the water is very likely.” should be removed, unless the authors perform at least a

computational docking experiment to prove their statement.

Author reply 2-2: We truly appreciate this valuable comment. In accordance with this recommendation, we rewrote the sentences at the Supplementary Information. We checked the glycerol binding in the Protein Data Bank. We found three of the crystal structures of the cryptochromes of 6X24, 5ZM0 (Figure 1) and 6PU0 (*Columba livia* CRY4 of pigeon). Like Figure 1, the CRY4 was also found to possess the relevant water binding site between $W_B(H)$ and $W_A(H)$, as shown in Figure L1. However, glycerol (GOL) molecules are not bound to these positions but different sites for three of the cryptochromes (Figure L2). Also, the distance between GOL and Flavin is 0.93 nm in Figure L1, denoting that the solvation energy is negligibly small contributed by GOL in the secondary RP state compared to the energy by the water molecule. Because it is widely accepted that glycerol molecules play a role to surround the protein surface area to protect the native protein structure as in DOI: 10.1021/bi900649t (Figure L2), we exclude the possibility of the glycerol binding to impact the solvation dynamics in Figure 4.

Figure L1. Crystal structure of *Columba livia* CRY4 of pigeon from the PDB code: 6PU0. GOL represents glycerol and is not located between W372 and W318. The distance between GOL and Flavin is 0.93 nm, denoting that the solvation energy is minor by GOL in the secondary RP state compared to the energy (0.4 eV) by the water solvation.

Figure L2. Crystal structure of *Chlamydomonas reinhardtii* (PDB code:5ZM0). GOL represents glycerol and is not located between W376 and W322. Both distances between GOL molecules and Flavin and between GOL and the trip-triad are too long (> 1 nm), denoting that the solvation energy contributions are negligible by GOL as in Figure L1.

About Trp inadvertent oxidation:

I agree about the Nernst equation, but probably the authors are forgetting that (1) there is a concentration factor in the Nerst equation; (2) Trp oxidation potential may significantly vary because of protein sorrounding, hydrogen bonding, dielectric constant, and pH. Please see

reference: <https://www.sciencedirect.com/science/article/pii/S0927776511002670>

Moreover, even a very small amount of free iron impurity (coming from ferricyanide) may represent a significant amount of catalyst in imidazole monooxygenation (as can be seen in the literature). Why the

authors do not provide a simple UV spectrum of the protein without any bound FAD that has been subjected to the same procedure ? This would immediately inform about the Trp(O) formation.

Author reply 2-3: We appreciate this comment. The standard reduction potential of potassium ferricyanide is 0.36 V vs. HNE. From the Nernst equation, this value is too low to oxidize even 1 % of the tryptophan residues with the oxidation potential of 1.0 V at pH = 8 (from J. AM. CHEM. SOC. 2005, 127, 3855-3863.) in the absence of the electrochemical potential from the electrode by the 5 mM potassium ferricyanide with a protein concentration of 10^{-4} M. However, the Trp(O) formation can occur at the environment of the electrochemical potential of 0.7 V as in the paper raised by this reviewer. Such a large electrochemical potential was not introduced in the preparation of the present sample. From the oxidation potential of 0.8 V in tyrosine at pH = 8 (from J. AM. CHEM. SOC. 2005, 127, 3855-3863), it is more likely that the oxidized state of tyrosine is produced rather than the tryptophan oxidization. To check the possible effect of impurity, we measured the UV-vis spectra (Figure L3). Oxidized forms of tryptophan and tyrosine were concluded to be very minor from the optical absorption bands (300-550 nm) based upon the literatures of Org. Biomol. Chem., 2014, 12, 3201–3210 and of Protein Science (2001), 10:735–740., as shown in Figure L3 and L4. The above explanations were added as Supplementary Figures 17 and 18 in the revised Supplementary Information.

Figure L3. UV-vis spectrum of *Xenopus laevis* CRY-DASH. Chromophore-bound protein with ferricyanide treatment (blue broken line), chromophore-removed protein without (black solid line) and with (red solid line) ferricyanide treatment proteins. Molar extension coefficient of chromophore-removed CRY-DASH was estimated by protein concentration by Bradford method, the molecular mass of the calculated His₆-tagged protein, and the absorption spectrum. The absorption spectrum of chromophore-bound protein was superimposed with FAD cofactor's extinction coefficient as $1.0 \times 10^4 \text{ M}^{-1} \text{ cm}^{-1}$ at 450 nm.

Oxindole absorption peak is reported to appear around 340 nm with an extinction coefficient of $1 \times 10^4 \text{ M}^{-1} \text{ cm}^{-1}$. (*Org. Biomol. Chem.* **12**, 3201-3210 (2014).) Oxidized tyrosine species from DOPA exhibits similar absorption bands around 300 nm and 470 nm. (*Protein. Sci.* **10**, 735-740 (2001) & *J. Biol. Chem.* **252**, 5729-5734, (1977)) Estimated from the absorbance difference around 330 nm (Inset), oxidized forms of tryptophan or tyrosine are thus 0.3 residues/molecule. A small absorption around 470 nm suggests existence of oxidized tyrosine with the extinction coefficient of $3.7 \times 10^3 \text{ M}^{-1} \text{ cm}^{-1}$ at 475 nm, indicating that 0.1 residues/molecule are oxidized in tyrosine. Possible candidates for oxidized tryptophan and tyrosine residues were shown in Figure L4, where 8 tryptophan and 12 tyrosine residues were exposed to the surface of the protein including W324 which corresponds W_C(H). (See below.) It is thus concluded that the oxidation of W_C(H) occurs less than 0.02 residues per molecule from a very small amount of free iron impurity. This minor (< 2 %) possibility of pre-oxidation in W_C(H)

is very consistent to the observation of the terminal CS state assigned to $FAD^{\cdot-} \cdots W_A(H) \cdots W_B(H) \cdots W_C(H)^{+\cdot}$ in Supplementary Figure 1 at the elevated temperature. This strongly supports the significant inhibition of the terminal CS event causing the water reorientation to $W_B(H)^{+\cdot}$ at the secondary CS state at 120 K.

(a)

X1_CRY_DASH	MCVPSRVIICLLRNDLRLHDNEVLHWAHRNADQIVPLCFDPRHYVGTHY	50
Synechocystis_CRY-DASH	MKHVPPTVLVWFRNDLRLHDHEPLHRAKLSGLAITAVCYDPRQFAQTHQ	50
	* . . . : :*****:* ** * . . . * . . :*:***:.. **	
X1_CRY_DASH	FNFPKTGPHRLKFLLESVRDLRITLKKKGSNLLLRGKPEEVIEDLVKQL	100
Synechocystis_CRY-DASH	G-FAKTGPWRSNFLQQSVQNLAESLQKVGKLLVTTGLPEQVIPQIAKQI	99
	*.**** * :** :***:* :*: * .:***: * **:* ** :.***:	
X1_CRY_DASH	GNVSAVTLHEEATKEETDVESAVKQACTRLGIKQTFWGSTLYHREDLPF	150
Synechocystis_CRY-DASH	N-AKTIYYHREVTQEELDVERNLVKQLTILGIEAKGYWGSTLCHPEDLPF	148
	. . . : : * . * . * ** * ** : : * * ** : : :***** * ** **	
X1_CRY_DASH	RHISSLPDVYITQFRKAVET-QGKVRPTFQMPDKLPLPSGLEEGVSPSHE	199
Synechocystis_CRY-DASH	S-IQDLPDLTKFRKDIEKKKISIRPCFFAPSQLLPSPNIKLELTAPPE	197
	* . *** : * : * ** : * . : . : * * * * * * . * : . * . *	
X1_CRY_DASH	DFDQQDPLTDPRTAFPCSGGESQALQRLEHYFWETNLVASYKDRNGLIG	249
Synechocystis_CRY-DASH	FFPQINFDHRSVLAFAQGGETAGLARLQDYFWHGDRDKDYKETRNGMVG	245
	* * : * * : . . . * ** : * * ** : * ** : . : . * ** : * ** : *	
X1_CRY_DASH	LDYSTKFAPWLLALGCVSPRYIEQIGKYEKERTANQSTYVWVIFELLWRDY	299
Synechocystis_CRY-DASH	ADYSSKFSPLALGCLSPRFIQEVKRYEQERSNDSTHWLIFELLWRDF	295
	::*****:***:***: . . . : : * ** : * : * ** : * ** : *	
X1_CRY_DASH	FRFVALKYGRRIFFLRGLQDKDIPWKRDPKLFDAWKEGRTGVPFVDANMR	349
Synechocystis_CRY-DASH	FRFVAQKYGKLNKLNKFNFPWQEDQVRFELWRSQGTGYPLVDANMR	345
	***** ** : . : * * * : * : * ** : * * : * . * : * ** : *	
X1_CRY_DASH	ELAMTGFMSNRGRQNVASFLTKDLGIDWRMGAEWFEYLLVDYDVCSNYGN	399
Synechocystis_CRY-DASH	ELNLTGFMSNRGRQNVASFLCKNLGIDWRMGAEWFEYCLIDYDVCSNYGN	395
	** . ***** * . ***** * ** * . ***** * . ***** * **	
X1_CRY_DASH	WLYSAGIGNDPRENKFNMIKQGLDYSGGDYIRLWVPELQQIKGGDAHT	449
Synechocystis_CRY-DASH	WNYTAGIGNDARDFRYFNIPKQSQQDPQGTYLRLHWLPELKNLPGDKIHQ	445
	* * :*****: * : * ** : * * . : * ** : * * : * * : * ** : *	
X1_CRY_DASH	PWALSNASLAHANLSLGETYPIVMAPEWSRHINQKPAGSWEKSARRGK	499
Synechocystis_CRY-DASH	PWLLSATEQKQWGVQLGVDYPRPCVNFHQSVEARRKIEQMGVIA-----	489
	** * * . . : . : * ** * * * : . . . : .	
X1_CRY_DASH	GPSHTPKQHKNRGIDFVFSRNKDV	523
Synechocystis_CRY-DASH	-----	489

Figure L4. Exploring candidates for oxidized tryptophan and tyrosine residues. (a) Comparison of the amino acid sequences of *Xenopus laevis* and *Synechocystis* CRY-DASHs. The amino acid residues corresponding to tryptophan and tyrosine in *Xenopus laevis* CRY-DASH are colored by yellow and green, respectively. (b) For the crystal structure of *Synechocystis* CRY-DASH (PDB ID: 1NP7), the residues corresponding to tryptophans (top) and tyrosines (bottom) in *Xenopus laevis* CRY-DASH are shown as sticks. The residues whose side chains are considered to be exposed on the surface are indicated by amino acid numbers of *Xenopus laevis* CRY-DASH.

Major 3

I thank the authors for taking into account my suggestion. I am very sorry they were not able to prepare the deuterium exchanged sample. It would have given a lot of information, given that they propose a phonon mechanism of coupling. I was not able to fully understand the problem of D₂O exchange they are facing, but I hope they could overcome these problems and publish their results. I think that an isotopic effect would be a clear evidence of the water tumbling effect.

Author reply 2-3: We are sorry for not being able to prepare the deuterium exchanged sample currently. However, we are planning to undergo TREPR experiments with a higher pH solution to examine an effect of binding of hydroxide to the tryptophan residues for the secondary and terminal CS states. This work together with the deuterium effect would ascertain the role of the single water molecule on the electron-tunneling in an unambiguous way and should be forthcoming.

However, as we emphasized repeatedly, the water solvation dynamics and the fluctuation dynamics are self-consistent at 120 K and 240 K, respectively. This is because no geometrical rearrangements were found on the secondary CS state and because the electronic coupling attenuation was not dramatical in the terminal CS state. We thus strongly believe that these novel findings are highly informative for the broad readership and are worthy of publication in *Communications Chemistry*.

Again, we truly thank this reviewer for above fruitful suggestions.

Overall I would accept the manuscript for publication after those minor revisions.

Reviewer #3 (Remarks to the Author):

The authors have updated the manuscript to meet my expectations. All

raised issues have been dealt with. While the manuscript could be published in its current form, I have one final suggestions: the arguments now presented with Figure S11 could/should be added to the main manuscript, because they are essential to the conceptual underpinning of the model suggested by the authors.

Author reply 3-1: We are pleased to see that this revised version is acceptable for the publication. Although the argument in Supplementary Figure 11 is essential to understand the present slow reorientation dynamics on the electron-transfer mechanism, this still supports the discussion of the simple concept shown in Figure 4b. We rather think that Supplementary Figure 11 may confuse the broad readers on the connection between the distribution of J and the solvent coordinate in Figure 4, when this Figure is in the main text. Thus, it is better that this supporting discussion is placed in the Supplementary Information, as the original version.